# Tight and Fast Bounds for Multi-Label Learning

**Yi-Fan Zhang** [1 2]   **Min-Ling Zhang** [2 3]

## Abstract

Commonly used evaluation metrics in multi-label learning all involve base loss functions, and the theoretical guarantees of multi-label learning often rely on the properties of base loss functions. Some recent theoretical works have used the Lipschitz continuity of base loss functions to prove the generalization bounds for multi-label learning, but the impact of the smoothness of base loss functions on the generalization bounds is completely unknown. In an attempt to make up for this gap in the generalization theory of multi-label learning, we develop some novel vector-contraction inequalities for smooth base loss functions and derive tight generalization bounds with no dependency on the number of labels, up to logarithmic terms. We then exploit local Rademacher complexity to develop some novel local vector-contraction inequalities for smooth base loss functions, which induce generalization bounds with a tighter dependency on the number of labels and a faster convergence rate with respect to the number of examples. In addition, we derive tight generalization bounds with no dependency on the number of labels, up to logarithmic terms, for Macro-Averaged AUC by exploiting the Lipschitz continuity and smoothness of base loss functions, respectively. Our state-of-the-art theoretical results provide general theoretical guarantees for the generalization of multi-label learning.

## 1. Introduction

Multi-label learning has attracted significant attention due to its ability to model real-world objects with rich semantics

[1]School of Cyber Science and Engineering, Southeast University, Nanjing 210096, China [2]Key Laboratory of Computer Network and Information Integration (Southeast University), Ministry of Education, China [3]School of Computer Science and Engineering, Southeast University, Nanjing 210096, China. Correspondence to: Min-Ling Zhang <zhangml@seu.edu.cn>.

*Proceedings of the 42nd International Conference on Machine Learning*, Vancouver, Canada. PMLR 267, 2025. Copyright 2025 by the author(s).

(Wang & Sukthankar, 2013; Sun et al., 2014; Nam et al., 2019), where each object is represented by a single instance associated with multiple class labels (Zhang & Zhou, 2014; Liu et al., 2022; Hang & Zhang, 2022). The goal of multi-label learning is to build proper classification models for objects assigned with multiple class labels simultaneously. In fact, multi-label scenarios widely exist in various real-word applications, such as text categorization (Schapire & Singer, 2000; Xun et al., 2020; Rubin et al., 2012), multi-media content annotation (Boutell et al., 2004; Cabral et al., 2011; Wu et al., 2014; You et al., 2020), bioinformatics (Barutçuoglu et al., 2006; Cesa-Bianchi et al., 2012; Chen et al., 2017) and other fields (Yu et al., 2005). Despite impressive advances in algorithms for multi-label learning, the theoretical analysis of generalization is still in its infancy.

Some recent theoretical works have made preliminary explorations into the generalization of multi-label learning, and have made some progress in the reduction of the dependency of the generalization bounds on the number of labels $c$ and the explicit introduction of label correlations in the generalization analysis (Wu & Zhu, 2020; Wu et al., 2021a;b; Zhang & Zhang, 2024a). However, from the perspective of learning theory, in some aspects, the generalization performance of multi-label learning is still completely unknown. These aspects that need to be explored include: 1) the establishment of faster convergence rates with respect to the number of examples $n$ of the generalization bounds, and 2) the impact of the smoothness of base losses on the generalization bounds. First, existing multi-label learning guarantees are centered around the concepts of Rademacher complexities (Bartlett & Mendelson, 2002; Koltchinskii & Panchenko, 2002), and the fastest convergence rate of the generalization bounds induced by Rademacher complexity is of the order $\widetilde{O}(1/\sqrt{n})$. The analysis based on local Rademacher complexity is capable of producing bounds with faster convergence rates than those obtained by Rademacher complexity (Bartlett et al., 2005; Koltchinskii, 2006), but it has not been well explored in multi-label learning. Hence, the development of general theoretical tools that can induce generalization bounds with faster convergence rates with respect to $n$ is a crucial open problem in the theory of multi-label learning. Second, although evaluation metrics with Lipschitz base losses have been extensively studied in existing theoretical work of multi-label learning, the bounds for evaluation met-

rics with smooth base losses remain completely unexplored, while the squared loss, as a representative of smooth base losses, is widely used by many multi-label learning methods (Huang et al., 2015; 2016; 2018; Weng et al., 2020; Yu & Zhang, 2022). Hence, generalization analysis is needed to provide general theoretical guarantees for multi-label learning methods with smooth base losses.

In this paper, we derive tight bounds with no dependency on $c$ and a faster convergence rate w.r.t. $n$ for multi-label learning, which connects the smoothness of base losses with the tighter and faster bounds. In addition, we improve the dependency of the bounds on $c$ to be independent for Macro-Averaged AUC both under Lipschitz and smooth conditions of base losses. Our theoretical analysis induces tighter and faster bounds and reveals the impact of smooth base losses on generalization. Major contributions of the paper include:

- We develop novel vector-contraction inequalities for smooth base losses, which induces tight bounds with no dependency on $c$ and provides general theoretical guarantees for multi-label learning methods with smooth base losses.

- We develop novel local vector-contraction inequalities for smooth base losses, which exploits local Rademacher complexity and can induce bounds with no dependency on $c$ and a faster rate w.r.t. $n$.

- We derive tight bounds with no dependency on $c$ for Macro-Averaged AUC with both Lipschitz and smooth base losses.

## 2. Related Work

We introduce the related work on generalization analysis for multi-label learning. Wu & Zhu (2020) first showed that one can obtain bounds of order $O(c/\sqrt{n})$ for multi-label learning with the typical vector-contraction inequality (Maurer, 2016) (i.e., for $\ell_2$ Lipschitz losses), which drew on the analysis of the relationship between the expectations of several evaluation metrics in (Dembczynski et al., 2010; 2012). Wu & Zhu (2020); Wu et al. (2021a) showed that the order of the bounds for Subset, Hamming and reweighted convex surrogate univariate loss can be improved to $O(\sqrt{c/n})$ when preserving the coupling among different components for kernel classes under the assumption of $\ell_2$ Lipschitz loss. Liu et al. (2018) also obtained a $O(\sqrt{c/n})$ bound for the dual set multi-label learning with the margin loss and kernel function classes. Zhang & Zhang (2024a) developed novel vector-contraction inequality for $\ell_2$ Lipschitz loss and derived a $\widetilde{O}(\sqrt{c/n})$ bound, which decouples the relationship among different components. Here we derive $\widetilde{O}(1/\sqrt{n})$ bounds for several evaluation metrics with smooth base losses, which improves the $\widetilde{O}(\sqrt{c/n})$ bounds for $\ell_2$ Lips-

chitz loss in (Zhang & Zhang, 2024a) by a $\sqrt{c}$ factor.

Wu et al. (2021b) derived a $O(\log(nc)/n\sigma)$ bound for $\ell_\infty$ Lipschitz loss with norm regularized kernel classes and the additional assumption that the regularizer is $\sigma$-strongly convex. Zhang & Zhang (2024a) derived a $\widetilde{O}(1/\sqrt{n})$ bound for $\ell_\infty$ Lipschitz loss, which decouples the relationship among different components. In addition, Yu et al. (2014) obtained a $O(1/\sqrt{n})$ bound for trace norm regularized linear function classes for the decomposable loss. Xu et al. (2016) used the local Rademacher complexity to derive a $\widetilde{O}(1/n)$ bound for trace norm regularized linear function classes with the assumption that the singular values of the weight matrix decay exponentially. Busa-Fekete et al. (2022) derived $\widetilde{O}(1/n)$ bounds for Hamming loss with KNN under sparsity, margin and smoothness assumptions and sharp bounds for Precision@$\kappa$ under margin and smoothness assumptions. Wu et al. (2023) obtained $O(1/\sqrt{n})$ bounds for Macro-Averaged AUC and gave thorough discussions about its relationships with the label-wise class imbalance. Here we derive sharp $\widetilde{O}(1/n)$ bounds for smooth base losses by exploiting local Rademacher complexity.

## 3. Preliminaries

Let $[n] := \{1, \ldots, n\}$ for any natural number $n$. In the context of multi-label learning, given a dataset $D = \{(\boldsymbol{x}_1, \boldsymbol{y}_1), \ldots, (\boldsymbol{x}_n, \boldsymbol{y}_n)\}$ with $n$ examples which are identically and independently distributed (i.i.d.) from a probability distribution $P$ on $\mathcal{X} \times \mathcal{Y}$, where $\mathcal{X} \subseteq \mathbb{R}^d$ denotes the $d$-dimensional input space and $\mathcal{Y}$ denotes the label space with $c$ class labels, $\boldsymbol{x} \in \mathcal{X}, \boldsymbol{y} \in \mathcal{Y} \subseteq \{-1, +1\}^c$, i.e., each $\boldsymbol{y} = (y_1, \ldots, y_c)$ is a binary vector and $y_j = 1$ ($y_j = -1$) denotes that the $j$-th label is relevant (irrelevant), $j \in [c]$. Multi-label learning aims to learn a multi-label prediction function $\boldsymbol{h} \in \mathcal{H} : \mathcal{X} \mapsto \{-1, +1\}^c$ which assigns each instance with a set of relevant labels.

### 3.1. Multi-Label Learning

A common strategy to solving multi-label learning is to learn a vector-valued function $\boldsymbol{f} = (f_1, \ldots, f_c) : \mathcal{X} \mapsto \mathbb{R}^c$ and derive the prediction function by a thresholding function which divides the label space into relevant and irrelevant label sets. We follow some definitions and notations in (Zhang & Zhang, 2024a). The general form of the prediction function for each label is $f_j(\boldsymbol{x}) = \langle \boldsymbol{w}_j, \phi_j(\boldsymbol{x}) \rangle$, where $\phi_j$ represents a nonlinear mapping. The function class of the multi-label learning is defined as follows:

$$\mathcal{F} = \{\boldsymbol{x} \mapsto \boldsymbol{f}(\boldsymbol{x}) : \boldsymbol{f}(\boldsymbol{x}) = (f_1(\boldsymbol{x}), \ldots, f_c(\boldsymbol{x})),$$
$$f_j(\boldsymbol{x}) = \boldsymbol{w}_j^\top \phi_j(\boldsymbol{x}), \boldsymbol{x} \in \mathcal{X}, j \in [c]$$
$$\boldsymbol{w} = (\boldsymbol{w}_1, \ldots, \boldsymbol{w}_c) \in \mathbb{R}^{d \times c}, \alpha(\boldsymbol{w}) \leq \Lambda,$$
$$\beta(\phi_j(\cdot)) \leq A, \Lambda > 0, A > 0\}, \qquad (1)$$

where $\alpha$ represents a functional that constrains weights, $\beta$ represents a functional that constrains nonlinear mappings.

Here we give some concrete examples as an explanation of the function class. For example, the DNN-based method named CLIF (Hang & Zhang, 2022), which proposes to learn label semantics and representations with specific discriminative properties for each class label in a collaborative way, can be expressed in the function class as $\phi_j(\boldsymbol{x}) = \sigma_{ReLU} \{W_5 \cdot [\sigma_{ReLU}(W_4\boldsymbol{x}) \odot \sigma_{sig}(W_3\psi(Y)_j)]\}$, the label embeddings $\psi(Y) := \sigma_{ReLU}(\tilde{A}\sigma_{ReLU}(\tilde{A}YW_1)W_2)$, where $\tilde{A}$ denote the normalized adjacency matrix with self-connections, $Y$ is the node feature matrix of the label graph, $\sigma_{ReLU}$ is the ReLU activation, $\sigma_{sig}$ is the sigmoid activation, $\odot$ is the Hadamard product, $W_i$ are the parameter metrices, $i \in [5]$. In addition, a class of multi-label methods based on the strategy of label-specific representation (Zhang & Zhang, 2024b), which facilitates the discrimination of each class label by tailoring its own representations, can be formalized in our function class. For example, the wrapped label-specific representation method (Yu & Zhang, 2022), which presents a kernelized Lasso-based framework with the constraint of pairwise label correlations for each class label, can be expressed in our function class, where $f_j$ is the kernelized linear model and the constraint $\alpha(\boldsymbol{w})$ is $\|\boldsymbol{w}_j\|_1 \leq \Lambda$ for any $j \in [c]$, and each label also has the property of sharing which is reflected by the additionally introduced constraint $\sum_i^c (1 - s_{ji})\boldsymbol{w}_j^\top \boldsymbol{w}_i \leq \tau$, where $s_{ji}$ is the cosine similarity between labels $y_j$ and $y_i$. Besides, the function class here is applicable to the typical Binary Relevance methods for multi-label learning, where different methods correspond to different nonlinear mappings $\phi_j$.

For any vector-valued function $\boldsymbol{f} : \mathcal{X} \mapsto \mathbb{R}^c$, the prediction quality on the example is measured by a loss function $\ell : \mathbb{R}^c \times \{-1, +1\}^c \mapsto \mathbb{R}_+$. The goal of learning is to find a hypothesis $\boldsymbol{f} \in \mathcal{F}$ with good generalization performance from the dataset $D$ by optimizing the loss $\ell$. We define the loss function space as $\mathcal{L} = \{\ell(\boldsymbol{f}(\boldsymbol{x}), \boldsymbol{y}) : \boldsymbol{f} \in \mathcal{F}\}$. The generalization performance is measured by the expected risk: $R(\ell(\boldsymbol{f})) = \mathbb{E}_{(\boldsymbol{x}, \boldsymbol{y}) \sim P}[\ell(\boldsymbol{f}(\boldsymbol{x}), \boldsymbol{y})]$. We denote the empirical risk w.r.t. the dataset $D$ as $\widehat{R}_D(\ell(\boldsymbol{f})) = \frac{1}{n}\sum_{i=1}^n \ell(\boldsymbol{f}(\boldsymbol{x}_i), \boldsymbol{y}_i)$. For simplicity, we slightly abuse $R(\boldsymbol{f})$ and $\widehat{R}_D(\boldsymbol{f})$ to represent $R(\ell(\boldsymbol{f}))$ and $\widehat{R}_D(\ell(\boldsymbol{f}))$.

The above definitions apply to Hamming, Subset and Ranking losses. However, the corresponding risk for Macro-Averaged AUC needs to be additionally defined since it involves the pairwise loss. The empirical risk w.r.t. Macro-Averaged AUC is defined as follows:

$$\widehat{R}_D(\boldsymbol{f}) \tag{2}$$
$$= \frac{1}{c}\sum_{j=1}^c \frac{1}{|X_j^+||X_j^-|}\sum_{\boldsymbol{x}_i \in X_j^+}\sum_{\boldsymbol{x}_i' \in X_j^-}\ell_{0/1}(f_j(\boldsymbol{x}_i) - f_j(\boldsymbol{x}_i')),$$

where $X_j^+ = \{\boldsymbol{x}_i \mid y_j = +1, i \in [n]\}$ ($X_j^- = \{\boldsymbol{x}_i' \mid y_j = -1, i \in [n]\}$) corresponds to the set of test instances that are relevant (irrelevant) to the $j$-th label. The expected risk w.r.t. Macro-Averaged AUC is defined as $R(\boldsymbol{f}) = \mathbb{E}_D[\widehat{R}_D(\boldsymbol{f})]$.

### 3.2. Related Evaluation Metrics

Many evaluation metrics are developed to measure the generalization performance of different multi-label learning methods. Here we focus on commonly used evaluation metrics, i.e., Hamming loss, Subset loss, Ranking loss and Macro-Averaged AUC. However, the corresponding losses in the above metrics are typically the 0-1 loss, which is hard to handle in practice. Hence, one usually consider their surrogate losses, which are defined as follows:

**Hamming Loss**: $\ell_H(\boldsymbol{f}(\boldsymbol{x}), \boldsymbol{y}) = \frac{1}{c}\sum_{j=1}^c \ell_b(f_j(\boldsymbol{x}), y_j)$,

where $\ell_b$ is the base convex surrogate loss.

**Subset Loss**: $\ell_S(\boldsymbol{f}(\boldsymbol{x}), \boldsymbol{y}) = \max_{j \in [c]}\{\ell_b(f_j(\boldsymbol{x}), y_j)\}$.

**Ranking Loss**:

$$\ell_R(\boldsymbol{f}(\boldsymbol{x}), \boldsymbol{y}) = \frac{1}{|Y^+||Y^-|}\sum_{p \in Y^+}\sum_{q \in Y^-}\ell_b(f_p(\boldsymbol{x}) - f_q(\boldsymbol{x})),$$

where $Y^+$ ($Y^-$) denotes the relevant (irrelevant) label index set induced by $\boldsymbol{y}$, and $|\cdot|$ denotes the cardinality of a set.

The induced surrogate loss for **Macro-Averaged AUC**:

$$\ell_M(\boldsymbol{f}(\boldsymbol{x}_i, \boldsymbol{x}_i'), \boldsymbol{y}) = \frac{1}{c}\sum_{j=1}^c \ell_b(f_j(\boldsymbol{x}_i) - f_j(\boldsymbol{x}_i')), \tag{3}$$

where $\boldsymbol{x}_i$ ($\boldsymbol{x}_i'$) corresponds to the instances that are relevant (irrelevant) to the $j$-th label.

The base surrogate loss functions $\ell_b$ involved in the above commonly used evaluation metrics can be various popular forms, such as Lipschitz loss functions with bounded derivative (including the hinge, margin, absolute-value, and logistic loss, etc) or smooth loss functions with the second derivative is bounded (including the squared, squared hinge, squared margin (Li et al., 2018) and smoothed ramp (Cortes et al., 2021) loss, etc). In this paper, we use $\ell$ to refer to the collective name for evaluation metrics such as Hamming loss, subset loss, and Ranking loss, etc, while $\ell_b$ is used specifically to refer to the base loss functions involved in the above commonly used evaluation metrics.

### 3.3. Related Complexity Measures

Here we introduce the related complexity measures involved in our theoretical results. The Rademacher and local Rademacher complexity are used to perform generalization analysis for multi-label learning.

**Definition 3.1** (Rademacher complexity). Let $\mathcal{F}$ be a class of real-valued functions mapping from $\mathcal{X}$ to $\mathbb{R}$. Let $D = $

$\{\boldsymbol{x}_1, \ldots, \boldsymbol{x}_n\}$ be a set with $n$ i.i.d. samples. The empirical **Rademacher complexity** over $\mathcal{F}$ is defined by

$$\hat{\Re}_D(\mathcal{F}) = \mathbb{E}_{\boldsymbol{\epsilon}} \left[ \sup_{f \in \mathcal{F}} \frac{1}{n} \sum_{i=1}^{n} \epsilon_i f(\boldsymbol{x}_i) \right],$$

where $\epsilon_1, \ldots, \epsilon_n$ are i.i.d. Rademacher random variables. In addition, we define the worst-case Rademacher complexity as $\tilde{\Re}_n(\mathcal{F}) = \sup_{D \in \mathcal{X}^n} \hat{\Re}_D(\mathcal{F})$.

The vector-valued function class $\mathcal{F}$ of multi-label learning makes traditional analysis methods developed for the Rademacher complexity of scalar-valued function class invalid. The typical vector-contraction inequality in (Maurer, 2016) shows that the Rademacher complexity of the class $\mathcal{F}$ composited with an $\ell_2$ Lipschitz function $h$ can be bounded by the maximum Rademacher complexity of the restrictions of the function class $\mathcal{F}$ along each coordinate and a factor $c$, and the induced bounds are not tight enough (Zhang & Zhang, 2024a). Therefore, we need new theoretical methods to convert the Rademacher complexity of a loss function space into the Rademacher complexity of a tractable scalar-valued function class for multi-label learning, especially for smooth base losses. The Rademacher complexity can be bounded by other scale-sensitive complexity measures, e.g. covering number and fat-shattering dimension (Srebro et al., 2010; Zhang & Zhang, 2023).

**Definition 3.2** (Covering number). Let $\mathcal{F}$ be a class of real-valued functions mapping from $\mathcal{X}$ to $\mathbb{R}$. Let $D = \{\boldsymbol{x}_1, \ldots, \boldsymbol{x}_n\}$ be a set with $n$ i.i.d. samples. For any $\epsilon > 0$ and $p \geq 1$, the empirical $\ell_p$ norm covering number $\mathcal{N}_p(\epsilon, \mathcal{F}, D)$ w.r.t. $D$ is defined as the minimal number $m$ of a collection of vectors $\boldsymbol{v}^1, \ldots, \boldsymbol{v}^m \in \mathbb{R}^n$ such that ($\boldsymbol{v}_i^j$ is the $i$-th component of the vector $\boldsymbol{v}^j$)

$$\left( \frac{1}{n} \sum_{i=1}^{n} |f(\boldsymbol{x}_i) - \boldsymbol{v}_i^j|^P \right)^{\frac{1}{p}} \leq \epsilon.$$

In this case, we call $\{\boldsymbol{v}^1, \ldots, \boldsymbol{v}^m\}$ an $(\epsilon, \ell_p)$-cover of $\mathcal{F}$ w.r.t. $D$. We also denote $\mathcal{N}_p(\epsilon, \mathcal{F}, n) = \sup_D \mathcal{N}_p(\epsilon, \mathcal{F}, D)$.

**Definition 3.3** (Fat-shattering dimension). Let $\mathcal{F}$ be a class of real-valued functions mapping from $\mathcal{X}$ to $\mathbb{R}$. We define the fat-shattering dimension $\text{fat}_\epsilon(\mathcal{F})$ at scale $\epsilon > 0$ as the largest $p \in \mathbb{N}$ such that there exist $p$ points $\boldsymbol{x}_1, \ldots, \boldsymbol{x}_p \in \mathcal{X}$ and witnesses $s_1, \ldots, s_p \in \mathbb{R}$ satisfying: for any $\delta_1, \ldots, \delta_p \in \{-1, +1\}$ there exists $f \in \mathcal{F}$ with

$$\delta_i \left( f(\boldsymbol{x}_i) - s_i \right) \geq \epsilon, \quad \forall i = 1, \ldots, p.$$

We also use the local Rademacher complexity to derive sharper bounds with a faster convergence rate with respect to the number of examples for multi-label learning methods with smooth base loss functions.

**Definition 3.4** (Local Rademacher complexity). For any $r > 0$, the expected **local Rademacher complexity** of the local loss function space $\mathcal{L}^r$ associated with a real-valued function class $\mathcal{F}$ is defined by

$$\hat{\Re}_D(\mathcal{L}^r) = \hat{\Re}_D(\{\ell(f(\boldsymbol{x}), y) : \ell \in \mathcal{L}, f \in \mathcal{F}, \widehat{R}_D(\ell(f)) \leq r\}),$$

where $\widehat{R}_D(\ell(f)) = \frac{1}{n} \sum_{i=1}^{n} \ell(f(\boldsymbol{x}_i), y_i)$.

## 4. Tighter Bounds for Smooth Base Losses

In this section, we first introduce the assumptions used in the generalization analysis, i.e., the assumption about the boundedness of functions and the assumption about the smoothness of base loss functions. Then, we develop some novel vector-contraction inequality for the Rademacher complexity of the loss function space $\mathcal{L}$ with smooth base loss functions. Finally, with these novel vector-contraction inequalities, we derive tight bounds for the general function class of multi-label learning with no dependency on the number of labels, up to logarithmic terms, which achieve the state of the art. **The proof sketches and detailed proofs of the theoretical results in this paper are provided in the appendix.**

**Assumption 4.1.** Assume that the loss function and the components of the vector-valued function are bounded: $\ell(\cdot, \cdot) \leq M$, $|f_j(\cdot)| \leq B$ for $j \in [c]$ where $M, B > 0$ are constants.

Assumption 4.1 is a pretty mild assumption. When we consider the function class (1) for multi-label learning, we will further use the constraint on weights ($\|\boldsymbol{w}_j\| \leq \Lambda$) and the constraint on nonlinear mappings ($\|\phi_j(\boldsymbol{x})\| \leq A$) to replace the boundedness of the components of $\boldsymbol{f}$, i.e., $B := \Lambda A$. When we analyze the generalization of the specific methods or models, we will further have $\|\phi_j(\boldsymbol{x})\| \leq \rho \|\boldsymbol{x}\|$ ($\phi_j$ is $\rho$-Lipschitz continuous) to take into account the differences of various methods or models. The differences in the Lipschitz constants between deep and shallow models are particularly obvious. The differences in the generalization of various methods or models are further reflected in the corresponding Lipschitz constants $\rho$. This work aims to explore capacity-based generalization bounds for smooth base loss functions and derive bounds that are weaker dependent on the number of labels, thereby providing general and effective theoretical guarantees for multi-label learning methods with smooth base loss functions. Therefore, here we only make general assumptions and do not specify specific methods or models, which is similar to Zhang & Zhang (2024a).

**Assumption 4.2.** Assume that the base loss functions $\ell_b$ involved in the commonly used evaluation metrics are $\gamma$-smooth, that is:

$$|\nabla \ell_b(s) - \nabla \ell_b(t)| \leq \gamma |s - t|.$$

Assumption 4.2 is a relatively mild condition. The squared, the squared hinge, the squared margin (Li et al., 2018) and the smoothed ramp (Cortes et al., 2021) losses all satisfy Assumption 4.2, especially the squared loss, which is widely used by many multi-label learning methods (Huang et al., 2015; 2016; 2018; Weng et al., 2020; Yu & Zhang, 2022).

We then show that the smoothness of base loss functions combined with the projection operator can induce novel tight vector-contraction inequalities. The projection operators are defined as $p_j : \mathbb{R}^c \mapsto \mathbb{R}$ for any $j \in [c]$ which project the $c$-dimensional vector onto the $j$-th coordinate. The projection function class is defined as $\mathcal{P}(\mathcal{F}) = \{(j, \boldsymbol{x}) \mapsto p_j(\boldsymbol{f}(\boldsymbol{x})) : p_j(\boldsymbol{f}(\boldsymbol{x})) = f_j(\boldsymbol{x}), \boldsymbol{f} \in \mathcal{F}, (j, \boldsymbol{x}) \in [c] \times \mathcal{X}\}$. With the above definitions, we develop some novel vector-contraction inequalities for smooth base loss functions, which show that the Rademacher complexities of loss function spaces can be bounded by the worst-case Rademacher complexity of the projection function class:

**Lemma 4.3.** *Let $\mathcal{F}$ be the class of multi-label learning defined by (1). Let Assumptions 4.1 and 4.2 hold. Given a dataset $D$ of size $n$. Then, we have*

$$\hat{\Re}_D(\mathcal{L}_H) \leq \frac{12M}{\sqrt{n}} + 192\sqrt{3c\gamma M}\widetilde{\Re}_{nc}(\mathcal{P}(\mathcal{F})) \times$$
$$(1 + \log_2(48en^2c^2\gamma M) \cdot \ln\frac{\sqrt{nM}}{\sqrt{\gamma}B}),$$

$$\hat{\Re}_D(\mathcal{L}_S) \leq \frac{12M}{\sqrt{n}} + 192\sqrt{3c\gamma M}\widetilde{\Re}_{nc}(\mathcal{P}(\mathcal{F})) \times$$
$$(1 + \log_2(48en^2c^2\gamma M) \cdot \ln\frac{\sqrt{nM}}{\sqrt{\gamma}B}),$$

$$\hat{\Re}_D(\mathcal{L}_R) \leq \frac{12M}{\sqrt{n}} + 384\sqrt{3c\gamma M}\widetilde{\Re}_{nc}(\mathcal{P}(\mathcal{F})) \times$$
$$(1 + \log_2(192en^2c^2\gamma M) \cdot \ln\frac{\sqrt{nM}}{\sqrt{\gamma}B}),$$

*where $\hat{\Re}_D(\mathcal{L}_H)$, $\hat{\Re}_D(\mathcal{L}_S)$ and $\hat{\Re}_D(\mathcal{L}_R)$ are the empirical Rademacher complexities of Hamming, Subset and Ranking loss space, respectively, and $\widetilde{\Re}_{nc}(\mathcal{P}(\mathcal{F}))$ is the worst-case Rademacher complexity of the projection function class.*

Then, we can derive the following tight bounds for multi-label learning with smooth base loss functions:

**Theorem 4.4.** *Let $\mathcal{F}$ be the class of multi-label learning defined by (1). Let Assumptions 4.1 and 4.2 hold. Given a dataset $D$ of size $n$. Then, for any $0 < \delta < 1$, with probability at least $1 - \delta$, the following holds for Hamming and Subset loss with smooth base losses and any $\boldsymbol{f} \in \mathcal{F}$:*

$$R(\boldsymbol{f}) \leq \widehat{R}_D(\boldsymbol{f}) + 3M\sqrt{\frac{\log\frac{2}{\delta}}{2n}} + \frac{24M}{\sqrt{n}} + \frac{384\sqrt{3\gamma M}B}{\sqrt{n}} \times$$
$$(1 + \log_2(48en^2c^2\gamma M) \cdot \ln\frac{\sqrt{nM}}{\sqrt{\gamma}B}),$$

*and the following holds for Ranking loss with smooth base losses and any $\boldsymbol{f} \in \mathcal{F}$:*

$$R(\boldsymbol{f}) \leq \widehat{R}_D(\boldsymbol{f}) + 3M\sqrt{\frac{\log\frac{2}{\delta}}{2n}} + \frac{24M}{\sqrt{n}} + \frac{768\sqrt{3\gamma M}B}{\sqrt{n}} \times$$
$$(1 + \log_2(192en^2c^2\gamma M) \cdot \ln\frac{\sqrt{nM}}{\sqrt{\gamma}B}).$$

*Remark* 4.5. The term $\widetilde{\Re}_{nc}(\mathcal{P}(\mathcal{F})) \leq B/\sqrt{nc}$ in Lemma 4.3, which makes the Rademacher complexity $\hat{\Re}_D(\mathcal{L}_H)$, $\hat{\Re}_D(\mathcal{L}_S)$ and $\hat{\Re}_D(\mathcal{L}_R)$ actually independent on $c$, up to logarithmic terms, and results in tighter bounds than the $O(c/\sqrt{n})$ bounds (Wu & Zhu, 2020; Wu et al., 2021a) and $\widetilde{O}(\sqrt{c}/\sqrt{n})$ bounds (Liu et al., 2018; Wu et al., 2021a; Zhang & Zhang, 2024a). Evaluation metrics with Lipschitz base loss functions have been extensively studied in previous work of multi-label learning. Wu & Zhu (2020) and Wu et al. (2021a) derived the bounds with a linear dependency on $c$ for $\ell_2$ norm Lipschitz losses, which comes from the typical vector-contraction lemma in (Maurer, 2016), and showed that the dependency of the bounds on $c$ can be improved to square-root by preserving the coupling among different components (i.e., with the constraint $\|\boldsymbol{w}\| \leq \Lambda$). Zhang & Zhang (2024a) improved the dependency of the bounds on $c$ from linear to square-root in the decoupling case for $\ell_2$ norm Lipschitz losses. Generalization bounds for evaluation metrics with smooth base losses are still completely unexplored. Our theoretical results here show that the dependency of the bounds in (Zhang & Zhang, 2024a) on $c$ can be improved from square-root to be independent by exploiting the smoothness of base loss functions, up to logarithmic terms, in the decoupling case. Our tight bounds with no dependency on $c$, up to logarithmic terms, can provide general theoretical guarantees for multi-label learning methods with smooth base loss functions (Huang et al., 2015; 2016; 2018; Weng et al., 2020; Yu & Zhang, 2022).

*Remark* 4.6. Zhang & Zhang (2024a) improved the dependency of the bounds on $c$ from linear to square-root in the decoupling case *for $\ell_2$ norm Lipschitz losses*. In fact, the results in Lemma 3.6 and Theorem 3.7 in (Zhang & Zhang, 2024a) require the introduction of an additional $\sqrt{c}$ factor, because the third step of the proof of Lemma 3.6 in (Zhang & Zhang, 2024a) ignores the $\sqrt{c}$ factor in the radius of the empirical $\ell_2$ cover of $\mathcal{P}(\mathcal{F})$, i.e., the process below equation (10) in the proof of Lemma 3.6 in (Zhang & Zhang, 2024a) should be modified as follows:

$$\hat{\Re}_D(\mathcal{L})$$
$$\leq \inf_{\alpha > 0}\left(4\alpha + \frac{12}{\sqrt{n}}\int_\alpha^M \sqrt{\log\mathcal{N}_2(\frac{\epsilon}{\mu\sqrt{c}}, \mathcal{P}(\mathcal{F}), [c] \times D)}d\epsilon\right)$$
$$\leq \inf_{\alpha > 0}\left(4\alpha + 48\sqrt{c}\mu\sqrt{c}\widetilde{\Re}_{nc}(\mathcal{P}(\mathcal{F}))\log^{\frac{1}{2}}(nc)\int_\alpha^M \epsilon^{-1}d\epsilon\right),$$

which will cause the bounds in Lemma 3.6 and Theorem 3.7 to be square-root dependent on $c$. We find that the square-root dependency of the bound in (Zhang & Zhang, 2024a) on $c$ is inevitable for $\ell_2$ norm Lipschitz losses, which essentially comes from the $\sqrt{c}$ factor in the radius of the empirical $\ell_2$ cover of the projection function class. We also find that the smoothness of the base loss function can eliminate the $\sqrt{c}$ factor in the radius of the empirical $\ell_2$ cover of the projection function class, so that the tight bound with no dependency on $c$, up to logarithmic terms, can be derived. In addition, the method based on Sudakov's minoration used in (Zhang & Zhang, 2024a) to upper bound the $\ell_2$ norm covering number of the projection function class is no longer applicable here. According to the above key points and proof ideas, for the bound with a square-root dependency on $c$ for $\ell_2$ Lipschitz loss in (Zhang & Zhang, 2024a), we consider the smoothness of the base loss and improve the bound by a factor of $\sqrt{c}$. In addition, the smoothness of the base loss combined with the local loss function space allows the development of novel local vector-contraction inequalities, which can induce bounds that not only have a faster convergence rate but also have a weaker dependency on $c$. Although for $\ell_2$ Lipschitz loss, the bounds in (Zhang & Zhang, 2024a) is only improved by a factor of $\sqrt{c}$, they are still the tightest results in multi-label learning with $\ell_2$ Lipschitz loss. In addition, for Hamming loss, its Lipschitz constant can induce the tight bounds with no dependency on $c$.

## 5. Faster Bounds for Smooth Base Losses

In this section, we first introduce the local Rademacher complexity used in the generalization analysis for multi-label learning. Then, we develop some novel local vector-contraction inequality for the Rademacher complexity of the local loss function space $\mathcal{L}^r$ with smooth base loss functions. Finally, with these novel vector-contraction inequalities, we derive sharper bounds for the general function class of multi-label learning with not only no dependency on the number of labels, up to logarithmic terms, but also a faster convergence rate with respect to the number of examples, which achieve the state of the art.

In fact, the Rademacher complexity considers the worst-case estimates of the complexity of the function class, it ignores the fact that the functions selected by the algorithm have a small error, and they are only a favorable small subset of the entire function class. As a result, the best convergence rate that can be obtained via the Rademacher complexity is at least of order $O(1/\sqrt{n})$. The local Rademacher complexity considers the Rademacher complexity of a small subset of the function class and is more reasonable as a complexity measure that yields faster convergence rates (Bartlett et al., 2005; Srebro et al., 2010; Reeve & Kaban, 2020; Li & Liu,

2021). Combined with the definition of the Rademacher complexity, we define the local Rademacher complexity for multi-label learning as follows:

**Definition 5.1** (Local Rademacher complexity for multi-label learning). Let $\mathcal{F}$ be the class of multi-label learning defined by (1). For any $r > 0$, the local function class of multi-label learning restricted by a functional is defined as:

$$\mathcal{F}^r := \{ \boldsymbol{x} \mapsto \boldsymbol{f}(\boldsymbol{x}) : \boldsymbol{f} \in \mathcal{F}, \widehat{R}_D(\ell(\boldsymbol{f})) \le r \},$$

where $\widehat{R}_D(\ell(\boldsymbol{f})) = \frac{1}{n} \sum_{i=1}^{n} \ell(\boldsymbol{f}(\boldsymbol{x}_i), \boldsymbol{y}_i)$. The local loss function space corresponding to $\mathcal{F}^r$ is defined as $\mathcal{L}^r := \{ \ell(\boldsymbol{f}(\boldsymbol{x}), \boldsymbol{y}) : \boldsymbol{f} \in \mathcal{F}^r \}$. The expected **local Rademacher complexity** of the local loss function space $\mathcal{L}^r$ associated with the local multi-label learning class $\mathcal{F}^r$ is defined by

$$\hat{\Re}_D(\mathcal{L}^r) = \hat{\Re}_D(\{ \ell(\boldsymbol{f}(\boldsymbol{x}), \boldsymbol{y}) : \boldsymbol{f} \in \mathcal{F}^r \}).$$

This means that a much smaller class $\mathcal{L}^r$ consisting of the functions $\boldsymbol{f}$ with a small error can yield sharper generalization bounds with a faster convergence rate.

With the assumption about the smoothness of base loss functions, we show that the local Rademacher complexity of the local loss function space $\mathcal{L}^r$ can be bounded by the worst-case Rademacher complexity of the projection function class. We develop the following novel local vector-contraction inequalities:

**Lemma 5.2.** *Let $\mathcal{F}$ be the class of multi-label learning defined by (1). Let Assumptions 4.1 and 4.2 hold. Given a dataset $D$ of size $n$. Then, we have*

$$\hat{\Re}_D(\mathcal{L}_H^r) \le \frac{48\sqrt{3\gamma r}(B+1)}{\sqrt{n}} + 48\sqrt{3c\gamma r}\widetilde{\Re}_{nc}(\mathcal{P}(\mathcal{F})) \times$$
$$(1 + 4\log_2(4Bn^{\frac{3}{2}}c)\log_2(\sqrt{n}B)),$$

$$\hat{\Re}_D(\mathcal{L}_S^r) \le \frac{48\sqrt{3\gamma r}(B+1)}{\sqrt{n}} + 48\sqrt{3c\gamma r}\widetilde{\Re}_{nc}(\mathcal{P}(\mathcal{F})) \times$$
$$(1 + 4\log_2(4Bn^{\frac{3}{2}}c)\log_2(\sqrt{n}B)),$$

$$\hat{\Re}_D(\mathcal{L}_R^r) \le \frac{96\sqrt{3\gamma r}(B+1)}{\sqrt{n}} + 96\sqrt{3c\gamma r}\widetilde{\Re}_{nc}(\mathcal{P}(\mathcal{F})) \times$$
$$(1 + 4\log_2(4Bn^{\frac{3}{2}}c)\log_2(\sqrt{n}B)).$$

*where $\hat{\Re}_D(\mathcal{L}_H^r)$, $\hat{\Re}_D(\mathcal{L}_S^r)$ and $\hat{\Re}_D(\mathcal{L}_R^r)$ are the empirical local Rademacher complexities of local Hamming, Subset and Ranking loss function spaces, respectively, and $\widetilde{\Re}_{nc}(\mathcal{P}(\mathcal{F}))$ is the worst-case Rademacher complexity of the projection function class.*

The challenge in using local Rademacher complexity to improve existing learning rates is to find the optimal radius that trades off the size of the subset we select and the associated complexity, which can be reduced to the calculation of the fixed point of a sub-root function. We introduce the definition and properties of the sub-root function as follows:

**Definition 5.3** (Sub-root function)**.** A function $\psi : [0, \infty)$ $\to [0, \infty)$ is sub-root if and only if it is non-decreasing and the function $r \mapsto \psi(r)/\sqrt{r}$ is non-increasing for $r > 0$.

**Lemma 5.4** (Lemma 3.2 in (Bartlett et al., 2005))**.** *If $\psi$ is a non-trivial sub-root function, then it is continuous on $[0, \infty]$, and the equation $\psi(r) = r$ has a unique positive solution $r^*$, which is known as the fixed point of $\psi$. Moreover, for any $r > 0$, it holds that $r > \psi(r)$ if and only if $r^* \leq r$.*

Then, with these novel local vector-contraction inequalities above and the properties of sub-root functions, we can derive the following sharper bounds for multi-label learning with smooth base loss functions:

**Theorem 5.5.** *Let $\mathcal{F}$ be the class of multi-label learning defined by (1). Let Assumptions 4.1 and 4.2 hold. Given a dataset $D$ of size $n$. Then, for any $0 < \delta < 1$, with probability at least $1 - \delta$, the following holds for Hamming and Subset loss with smooth base losses and any $\boldsymbol{f} \in \mathcal{F}$:*

$$R(\boldsymbol{f}) \leq 2\widehat{R}_D(\boldsymbol{f}) + \frac{bM(\log \frac{1}{\delta} + 6\log\log n)}{n} + \frac{3 \cdot 48^2 a\gamma}{n} \times$$
$$(B+1)^2 \left(1 + (1 + 4\log_2(4Bn^{\frac{3}{2}}c)\log_2(\sqrt{n}B))\right)^2,$$

*and the following holds for Ranking loss with smooth base losses and any $\boldsymbol{f} \in \mathcal{F}$:*

$$R(\boldsymbol{f}) \leq 2\widehat{R}_D(\boldsymbol{f}) + \frac{bM(\log \frac{1}{\delta} + 6\log\log n)}{n} + \frac{3 \cdot 96^2 a\gamma}{n} \times$$
$$(B+1)^2 \left(1 + (1 + 4\log_2(4Bn^{\frac{3}{2}}c)\log_2(\sqrt{n}B))\right)^2,$$

*where $a = 106$, $b = 48$.*

*Remark* 5.6. The term $\tilde{\Re}_{nc}(\mathcal{P}(\mathcal{F})) \leq B/\sqrt{nc}$ in Lemma 5.2, which makes the local Rademacher complexity $\hat{\Re}_D(\mathcal{L}_H^r)$, $\hat{\Re}_D(\mathcal{L}_S^r)$ and $\hat{\Re}_D^r(\mathcal{L}_R)$ actually independent on $c$, up to logarithmic terms, and results in sharper bounds than the $\widetilde{O}(1/\sqrt{n})$ bounds in Theorem 4.4 with a faster convergence rate $\widetilde{O}(1/n)$. Theorem 5.5 is the most advanced theoretical result for multi-label learning so far. Although Xu et al. (2016) also derived a generalization bound of order $\widetilde{O}(1/n)$ by using the local Rademacher complexity, this bound is accompanied by the trace norm regularized linear function classes and the assumption that the singular values of the weight matrix decay exponentially. These strong assumptions lead to their bound being completely independent of the number of labels. On the one hand, the trace norm regularizer actually preserves the coupling among different components. Wu & Zhu (2020) first revealed that preserving the coupling can improve the dependency of the bound on the number of labels by a factor of $\sqrt{c}$ in multi-label learning. Zhang & Zhang (2024a) revealed that the trace norm regularizer actually corresponds to the case of preserving the coupling. In addition, the analysis of the linear prediction function class in (Xu et al., 2016) rather than

the loss function space associated with the vector-valued prediction function class in (Wu & Zhu, 2020; Wu et al., 2021a; Zhang & Zhang, 2024a) leads to the improvement of the dependency on the number of labels by a factor of $\sqrt{c}$. Hence, under these strong assumptions, the bound in (Xu et al., 2016) is independent of the number of labels. On the other hand, the faster convergence rate $\widetilde{O}(1/n)$ of the bound in (Xu et al., 2016) comes from the assumption that the singular values of the weight matrix decay exponentially. It is not obvious to satisfy such an assumption and requires careful design of the algorithm. Hence, the bound in (Xu et al., 2016) is not general and is heavily specific to their proposed algorithm. Our bounds here are not only independent of the number of labels, but also have a faster convergence rate. The only assumption is that the base loss functions involved in evaluation metrics are smooth, which is very mild in multi-label learning since many multi-label learning methods use smooth base loss functions (Huang et al., 2015; 2016; 2018; Weng et al., 2020; Yu & Zhang, 2022). Hence, our bounds can provide general theoretical guarantees for multi-label learning methods and datasets, and also explain the good generalization performance of multi-label learning methods with smooth base loss functions.

*Remark* 5.7. Busa-Fekete et al. (2022) also derived tight bounds with a logarithmic dependency on $c$ for Hamming loss with KNN under the smoothness assumption of the regression function and multi-label margin and sparsity assumptions and also derived tight bounds with a logarithmic dependency on $c$ for Precision@$\kappa$ under the margin condition and the smoothness assumption. The margin condition ensures that the obtained bounds with a faster convergence rate. In our work, the local loss function space is the key to obtaining bounds with a faster convergence rate. The smoothness condition with respect to the $\ell_\infty$ norm in (Busa-Fekete et al., 2022) is a variant of Holder-continuity, which is weaker than the standard smoothness condition. We also find that the $\ell_\infty$ norm has a positive effect on obtaining tight bound with a weaker dependency on $c$, i.e., tight bounds with a logarithmic dependency on $c$ can be derived for $\ell_\infty$ Lipschitz losses. However, how to improve the convergence rate of the bounds for Lipschitz losses is still an open problem, which we will further explore in future work.

## 6. Tighter Bounds for Macro-Averaged AUC

In this section, we derive some tight bounds for Macro-Averaged AUC. Specifically, we first exploit the Lipschitz continuity of base losses to develop a novel vector-contraction inequality, which can induce tight bounds with no dependency on $c$ for Macro-Averaged AUC. Then, we exploit the smoothness of base losses to develop a novel vector-contraction inequality, which can also induce tight bounds with no dependency on $c$ for Macro-Averaged AUC. These bounds achieve the state of the art and also reveal

the relationship between Macro-Averaged AUC and class-imbalance (Wu et al., 2023; Zhang & Zhang, 2024a).

We first introduce the following assumption about the Lipschitz continuity with respect to the $\ell_\infty$ norm:

**Assumption 6.1.** Assume that the loss function $\ell$ is $\rho$-Lipschitz continuous w.r.t. the $\ell_\infty$ norm, that is:

$$\left|\ell(\boldsymbol{f}(\boldsymbol{x}), \cdot) - \ell(\boldsymbol{f}'(\boldsymbol{x}), \cdot)\right| \leq \rho \left\|\boldsymbol{f}(\boldsymbol{x}) - \boldsymbol{f}'(\boldsymbol{x})\right\|_\infty,$$

where $\rho > 0$, $\|\boldsymbol{t}\|_\infty = \max_{j \in [c]} |t_j|$ for $\boldsymbol{t} = (t_1, \ldots, t_c)$.

The Lipschitz continuity with respect to the $\ell_\infty$ norm has been considered in several literature (Foster & Rakhlin, 2019; Lei et al., 2019; Wu et al., 2021b; Zhang & Zhang, 2024a). Zhang & Zhang (2024a) showed that when the base loss functions are $\rho$-Lipschitz continuous, the induced surrogate loss for Macro-Averaged AUC in Subsection 3.2 is $\rho$-Lipschitz continuous. Next, we exploit the Lipschitz continuity of base loss functions to develop a novel vector-contraction inequality for $\ell_\infty$ norm Lipschitz loss:

**Lemma 6.2.** *Let $\mathcal{F}$ be the class of multi-label learning defined by (1). Let Assumptions 4.1 and 6.1 hold. Given a dataset $D$ of size $n$. Then, we have*

$$\hat{\Re}_D(\mathcal{L}) \leq \frac{12M}{\sqrt{n}} + 96\rho\sqrt{c}\widetilde{\Re}_{nc}(\mathcal{P}(\mathcal{F})) \times$$
$$(1 + \log_2(4en^2c^2\rho^2) \cdot \ln \frac{M\sqrt{n}}{\rho B}),$$

*where $\widetilde{\Re}_{nc}(\mathcal{P}(\mathcal{F}))$ is the worst-case Rademacher complexity of the projection function class.*

We follow the definitions and notations of Macro-Averaged AUC in (Zhang & Zhang, 2024a). Let $p_j$ be the probability that the samples are relevant to the $j$-th label. $\mathcal{D}_j^+$ denotes the conditional distribution of the samples over $\mathcal{X}$ given that the samples are relevant to the $j$-th label, and $\mathcal{D}_j^-$ denotes the conditional distribution of the samples over $\mathcal{X}$ given that the samples are irrelevant to the $j$-th label. We denote $\left|X_j^+\right|$ and $\left|X_j^-\right|$ in (2) as $s_j$ and $t_j$, $s_j + t_j = n$ for any $j \in [c]$, and denote the number of disjoint positive and negative sample pairs for the $j$-th label as $r_j = \min\{s_j, t_j\}$. We construct the set of positive and negative sample pairs $(\boldsymbol{x}_i^{j+}, \boldsymbol{x}_i^{j-})$ $(i \in [r_j])$ by matching the samples from the set of the samples that are relevant to the $j$-th label with the samples from the set of the samples that are irrelevant to the $j$-th label until one of the sets of the positive samples and the negative samples exhausts its available samples for selection. We denote the set of i.i.d disjoint positive and negative sample pairs for the $j$-th class label as $D_j$, and $|D_j| = r_j$. Then, with these definitions and permutations in U-process (Clémençon et al., 2008; Zhang & Zhang, 2024a), we derive the tight bound for Macro-Averaged AUC with Lipschitz base losses as follows:

**Theorem 6.3.** *Let $\mathcal{F}$ be the class of multi-label learning defined by (1). Let Assumptions 4.1 and 6.1 hold. Given a dataset $D$ of size $n$. Then, for Macro-Averaged AUC with Lipschitz base losses, for any $0 < \delta < 1$, with probability at least $1 - \delta$, the following holds for any $\boldsymbol{f} \in \mathcal{F}$:*

$$R(\boldsymbol{f}) \leq \widehat{R}_D(\boldsymbol{f}) + \frac{24\sqrt{2}M}{\sqrt{r_0}} + 18M\sqrt{\frac{\ln\frac{4}{\delta}}{r_0}} + \frac{384\sqrt{2}\rho B}{\sqrt{r_0}} \times$$
$$(1 + \log_2(4er_0^2c^2\rho^2) \cdot \ln \frac{M\sqrt{r_0}}{\rho B}),$$

*where $r_0 = n \min\{\min_j p_j, \min_j(1 - p_j)\}$.*

*Remark* 6.4. We define the empirical Rademacher complexity of a loss function space associated with the multi-label learning class $\mathcal{F}$ over the set of i.i.d disjoint positive and negative sample pairs for the $j$-th label as $\hat{\Re}_{D_j}(\mathcal{L}_M) = \mathbb{E}_\epsilon \left[\sup_{\boldsymbol{f} \in \mathcal{F}} \frac{1}{r_j} \sum_{i=1}^{r_j} \frac{1}{c} \sum_{j=1}^c \epsilon_i \ell_b\left(f_j(\boldsymbol{x}_i^{j+}) - f_j(\boldsymbol{x}_i^{j-})\right)\right]$, by using the U-process technique, we find that the generalization error can be bounded by $\hat{\Re}_{D_j}(\mathcal{L}_M)$. Since $r_j$ is random, we incorporate an additional Chernoff-type argument to obtain a bound that does not involve any random quantities, and upper bound the generalization error by $\hat{\Re}_{D_0}(\mathcal{L}_M) \leq \frac{12M}{\sqrt{r_0}} + \frac{192\rho B}{\sqrt{r_0}}(1 + \log_2(4er_0^2c^2\rho^2) \cdot \ln \frac{M\sqrt{r_0}}{\rho B})$.

*Remark* 6.5. The $\widetilde{O}(1/\sqrt{r_0})$ bound with no dependency on $c$ in Theorem 6.3 improve the $\widetilde{O}(\sqrt{c/r_0})$ bound in (Zhang & Zhang, 2024a) by a factor of $\sqrt{c}$. When class-imbalance is more serious, $r_0$ will be smaller, the bound for Macro-Averaged AUC will be looser. The $\widetilde{O}(\sqrt{c/r_0})$ bound in (Zhang & Zhang, 2024a) is more likely to be vacuous (i.e., $> 1$) since class-imbalance makes $c > r_0$ more likely to occur. Hence, the $\widetilde{O}(1/\sqrt{r_0})$ bound here, rather than the $\widetilde{O}(\sqrt{c/r_0})$ bound in (Zhang & Zhang, 2024a), can provide a general theoretical guarantee for Macro-Averaged AUC.

Then, we develop a novel vector-contraction inequality for Macro-Averaged AUC with smooth base loss functions:

**Lemma 6.6.** *Let $\mathcal{F}$ be the class of multi-label learning defined by (1). Let Assumptions 4.1 and 4.2 hold. Given a dataset $D$ of size $n$. Then, we have*

$$\hat{\Re}_D(\mathcal{L}_M) \leq \frac{12M}{\sqrt{n}} + 192\sqrt{3c\gamma M}\widetilde{\Re}_{nc}(\mathcal{P}(\mathcal{F})) \times$$
$$(1 + \log_2(48en^2c^2\gamma M) \cdot \ln \frac{\sqrt{nM}}{\sqrt{\gamma}B}),$$

*where $\hat{\Re}_D(\mathcal{L}_M)$ is the empirical Rademacher complexities of the loss function space of the induced surrogate loss for Macro-Averaged AUC, and $\widetilde{\Re}_{nc}(\mathcal{P}(\mathcal{F}))$ is the worst-case Rademacher complexity of the projection function class.*

With the above vector-contraction inequality, we derive tight bound for Macro-Averaged AUC with smooth base losses:

**Theorem 6.7.** *Let $\mathcal{F}$ be the class of multi-label learning defined by (1). Let Assumptions 4.1 and 4.2 hold. Given a dataset $D$ of size $n$. Then, for any $0 < \delta < 1$, with probability at least $1 - \delta$, the following holds for Macro-Averaged AUC with smooth base losses and any $\boldsymbol{f} \in \mathcal{F}$:*

$$R(\boldsymbol{f}) \leq \widehat{R}_D(\boldsymbol{f}) + \frac{24\sqrt{2}M}{\sqrt{r_0}} + 18M\sqrt{\frac{\ln\frac{4}{\delta}}{r_0}} + \frac{768\sqrt{6\gamma M}B}{\sqrt{r_0}}(1 + \log_2(48er_0^2c^2\gamma M) \cdot \ln\frac{\sqrt{r_0 M}}{\sqrt{\gamma}B}),$$

*where $r_0 = n\min\{\min_j p_j, \min_j(1 - p_j)\}$.*

*Remark* 6.8. We derive tight $\widetilde{O}(1/\sqrt{r_0})$ bounds for Macro-Averaged AUC by exploiting both Lipschitz continuity and smoothness of base losses. Wu et al. (2023) also obtained bounds with no dependency on $c$ for Macro-Averaged AUC by their proposed new McDiarmid-type inequality, and provided thorough analysis on its relationships with the label-wise class-imbalance. However, we focus on studying the impact of the properties of base losses on the bound, and are committed to developing new analysis methods and theoretical tools under Lipschitz and smooth conditions of base losses to reduce the dependency of the bound on $c$.

## 7. Discussion

When there is some type of label correlation between the labels of the dataset, the label distribution may satisfy some potential constraints, which may correspond to some additional assumptions such as sparsity assumptions or norm regularization constraints. Therefore, when dealing with these specific problems, we need to introduce some additional assumptions to adjust our analysis and explicitly introduce these potential label correlations into the generalization analysis. Different types of label correlations have an important impact on the generalization analysis. How to explicitly introduce them in generalization analysis is a crucial open problem, and we will further explore related work in the future. When dealing with large-scale datasets, in practice, one often consider introducing specific strategies in the label space to deal with extremely large number of labels. In generalization analysis, it is shown that introducing effective and general specific strategies is not only an important open problem in theory, but also extremely challenging in practice. Therefore, it is indeed necessary to further introduce effective assumptions to better explicitly analyze the role of key factors in generalization that can effectively deal with challenging problems with large number of labels and large-scale datasets. In summary, effective analysis for more specific problems requires further explicit introduction of valid assumptions to reveal the impact of these setting-dependent factors on the bound. For example, for high-dimensional sparse data, one may need to introduce sparsity assumptions into the analysis, thereby inducing bounds that are weakly dependent on the sparsity rate and the number of key labels.

Our theoretical results do not cover the case where the base loss is the cross-entropy loss, mainly because the cross-entropy loss is not bounded. On the smoothness of the cross-entropy loss, when the model is a linear classifier, the smoothness of the cross-entropy loss can be achieved by the boundedness of the input, but from the perspective of model capacity, such a result is not general. The function class here involves general functions (i.e., nonlinear mappings $\phi_j$), so for nonlinear models, the smoothness of the cross-entropy loss involves not only the boundedness of the first-order gradient of the model function but also the boundedness of the second-order derivative. For deep networks, changes in parameters may cause drastic changes in the second-order derivative, resulting in the norm of the second-order derivative being unbounded. Hence, the smoothness of the cross-entropy is often difficult to hold. However, the Lipschitz continuity of the cross-entropy is often established, and the boundedness of the gradient of the model function is guaranteed by various strategies in practice, such as input normalization, weight initialization, gradient clipping, and regularization. This implies the need to develop new theories and analytical methods for unbounded base losses. Under the new theoretical analysis method, tight bounds with no dependency on $c$ for cross-entropy loss can be obtained using its Lipschitz continuity, but improving the convergence rate of its bound with respect to the number of examples is still an urgent open problem to be solved. In the future, we will further study the bounds with a faster convergence rate for unbounded and Lipschitz base losses.

## 8. Conclusion

In this paper, we develop novel vector-contraction inequalities for smooth base losses, which induces tight $\widetilde{O}(1/\sqrt{n})$ bounds with no dependency on $c$. We also derive $\widetilde{O}(1/n)$ bounds with no dependency on $c$ and a faster rate w.r.t. $n$ by exploiting local Rademacher complexity. In addition, we derive tight bounds with no dependency on $c$ for Macro-Averaged AUC with both Lipschitz and smooth base losses.

In future work, we will consider experimental verification from two aspects. On the one hand, verify whether the functions selected by the algorithm have a small error and whether the generalization performance of the small error functions is better. On the other hand, verify whether the smoothness of the model function can be guaranteed by some regularization, and explore which regularization-induced inductive biases are more effective for generalization in practice. We will also develop new theoretical results to investigate the lower bound for multi-label learning to test whether our bounds here are optimal.

## Acknowledgements

The authors wish to thank the anonymous reviewers for their helpful comments and suggestions. This work was supported by the National Science Foundation of China (62225602).

## Impact Statement

This paper presents work whose goal is to advance the field of Machine Learning. There are many potential societal consequences of our work, none which we feel must be specifically highlighted here.

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

# A. Appendix

## A.1. Appendix Outline

In the appendix, we give the detailed proofs of our theoretical results in the main paper. Our main proofs include:

- The tight vector-contraction inequalities for multi-label learning with smooth base losses (Lemma 4.3).

- The tight bounds for multi-label learning with smooth base losses (Theorem 4.4).

- The local vector-contraction inequalities for multi-label learning with smooth base losses (Lemma 5.2)

- The faster bounds for multi-label learning with smooth base losses (Theorem 5.5)

- The novel vector-contraction inequality for $\ell_\infty$ norm Lipschitz loss (Lemma 6.2)

- The tight bound for Macro-Averaged AUC with Lipschitz base losses (Theorem 6.3)

- The novel vector-contraction inequality for Macro-Averaged AUC with smooth base losses (Lemma 6.6)

- The tight bound for Macro-Averaged AUC with smooth base losses (Theorem 6.7)

## A.2. Preliminaries

### A.2.1. The Bound for the loss function space

According to McDiarmid's inequality (McDiarmid et al., 1989) and the symmetrization technique (e.g., Theorem 4.4 in (Mohri et al., 2018)), it is easy to obtain that for any training dataset $D = \{(\boldsymbol{x}_i, \boldsymbol{y}_i) : i \in [n]\}$, $\ell(\cdot, \cdot) \leq M$, with probability at least $1 - \delta$, the following holds:

$$R(\ell(\boldsymbol{f})) \leq \widehat{R}_D(\ell(\boldsymbol{f})) + 2\hat{\Re}_D(\mathcal{L}) + 3M\sqrt{\frac{\log \frac{2}{\delta}}{2n}}. \tag{4}$$

## A.3. Tighter Bounds for Smooth Base Losses

### A.3.1. Proof of Lemma 4.3

**Proof Sketch**: For Hamming loss, according to the smoothness of base loss functions, we first derive the relationship between the empirical $\ell_2$ norm covering number $\mathcal{N}_2(\epsilon, \mathcal{L}_H, D)$ of the loss space $\mathcal{L}_H$ and the empirical $\ell_\infty$ norm covering number $\mathcal{N}_\infty(\epsilon, \mathcal{P}(\mathcal{F}), [c] \times D)$ of the projection function class $\mathcal{P}(\mathcal{F})$. Then, we show that the empirical $\ell_\infty$ norm covering number $\mathcal{N}_\infty(\epsilon, \mathcal{P}(\mathcal{F}), [c] \times D)$ of the projection function class $\mathcal{P}(\mathcal{F})$ can be bounded by the fat-shattering dimension, and the fat-shattering dimension can be bounded by the worst-case Rademacher complexity of the projection function class $\mathcal{P}(\mathcal{F})$. Finally, combining the above results and the refined Dudley's entropy integral inequality, the Rademacher complexity of the loss function space $\mathcal{L}_H$ can be bounded by the worst-case Rademacher complexity of the projection function class $\mathcal{P}(\mathcal{F})$, and the desired bound can be derived. For Subset loss and Ranking loss, we derive the relationships between the empirical $\ell_2$ norm covering number $\mathcal{N}_2(\epsilon, \mathcal{L}_S, D)$, $\mathcal{N}_2(\epsilon, \mathcal{L}_R, D)$ and the empirical $\ell_\infty$ norm covering number $\mathcal{N}_\infty(\epsilon, \mathcal{P}(\mathcal{F}), [c] \times D)$, respectively. Then, using the similar technique to the tight vector-contraction inequality for Hamming loss, the desired bounds for Subset and Ranking loss can be derived.

We first introduce the following lemmas:

**Lemma A.1** (Lemma A.1 in (Srebro et al., 2010)). *For any $H$-smooth non-negative function $f : \mathbb{R} \mapsto \mathbb{R}$ and any $t, r \in \mathbb{R}$ we have that*

$$(f(t) - f(r))^2 \leq 6H(f(t) + f(r))(t - r)^2.$$

**Lemma A.2** (Khintchine-Kahane inequality (Lust-Piquard & Pisier, 1991)). *Let $\boldsymbol{v}_1, \ldots, \boldsymbol{v}_n \in \mathcal{H}$, where $\mathcal{H}$ is a Hilbert space with $\|\cdot\|$ being the associated $p$-th norm. Let $\epsilon_1, \ldots, \epsilon_n$ be a sequence of independent Rademacher variables. Then, for any $p \geq 1$ there holds*

$$\min(\sqrt{p-1}, 1) \left[\sum_{i=1}^n \|\boldsymbol{v}_i\|^2\right]^{\frac{1}{2}} \leq \left[\mathbb{E}_{\boldsymbol{\epsilon}} \left\|\sum_{i=1}^n \epsilon_i \boldsymbol{v}_i\right\|^p\right]^{\frac{1}{p}} \leq \max(\sqrt{p-1}, 1) \left[\sum_{i=1}^n \|\boldsymbol{v}_i\|^2\right]^{\frac{1}{2}},$$

*and*

$$\mathbb{E}_{\boldsymbol{\epsilon}} \left\| \sum_{i=1}^{n} \epsilon_i \boldsymbol{v}_i \right\| \geq 2^{-\frac{1}{2}} \left[ \sum_{i=1}^{n} \|\boldsymbol{v}_i\|^2 \right]^{\frac{1}{2}}.$$

**Lemma A.3** (Lemma A.2 in (Srebro et al., 2010)). *For any function class $\mathcal{F}$, any $S$ with a finite sample of size $n$ and any $\epsilon > \hat{\mathfrak{R}}_S(\mathcal{F})$, we have that*

$$\mathrm{fat}_\epsilon(\mathcal{F}) \leq \frac{4n\hat{\mathfrak{R}}_S^2(\mathcal{F})}{\epsilon^2}.$$

**Lemma A.4** (Theorem 12.8 in (Anthony & Bartlett, 2009), (Lei et al., 2023)). *If any function in class $\mathcal{F}$ takes values in $[-B, B]$, then for any $S$ with a finite sample of size $n$, any $\epsilon > 0$ with $\mathrm{fat}_\epsilon(\mathcal{F}) < n$, we have*

$$\log \mathcal{N}_\infty (\epsilon, \mathcal{F}, S) \leq 1 + d \log_2 \frac{4eBn}{d\epsilon} \log \frac{4nB^2}{\epsilon^2},$$

*where $d = \mathrm{fat}_{\epsilon/4}(\mathcal{F})$.*

**Lemma A.5** (Refined Dudley's entropy integral inequality, Lemma C.5 in (Zhang & Zhang, 2024a)). *Let $\mathcal{F}$ be a real-valued function class with $f \leq B$, $f \in \mathcal{F}$, $B > 0$, and assume that $0 \in \mathcal{F}$. Let $S$ be a finite sample of size $n$. For any $2 \leq p \leq \infty$, we have the following relationship between the Rademacher complexity $\hat{\mathfrak{R}}_S(\mathcal{F})$ and the covering number $\mathcal{N}_p(\epsilon, \mathcal{F}, S)$.*

$$\hat{\mathfrak{R}}_S(\mathcal{F}) \leq \inf_{\alpha > 0} \left( 4\alpha + \frac{12}{\sqrt{n}} \int_\alpha^B \sqrt{\log \mathcal{N}_p(\epsilon, \mathcal{F}, S)} d\epsilon \right).$$

First, we derive the tight vector-contraction inequality for Hamming loss with smooth base loss functions.

**Step 1**: We first derive the relationship between the empirical $\ell_2$ norm covering number $\mathcal{N}_2(\epsilon, \mathcal{L}_H, D)$ and the empirical $\ell_\infty$ norm covering number $\mathcal{N}_\infty(\epsilon, \mathcal{P}(\mathcal{F}), [c] \times D)$ by the smoothness of base loss functions.

For the dataset $D = \{(\boldsymbol{x}_1, \boldsymbol{y}_1), \ldots, (\boldsymbol{x}_n, \boldsymbol{y}_n)\}$ with $n$ i.i.d. examples:

$$\sqrt{\frac{1}{n} \sum_{i=1}^{n} \left( \ell_H(\boldsymbol{f}(\boldsymbol{x}_i), \boldsymbol{y}_i) - \ell_H(\boldsymbol{f}'(\boldsymbol{x}_i), \boldsymbol{y}_i) \right)^2}$$

$$= \sqrt{\frac{1}{n} \sum_{i=1}^{n} \left( \frac{1}{c} \sum_{j=1}^{c} \ell_b(f_j(\boldsymbol{x}_i), y_{ij}) - \frac{1}{c} \sum_{j=1}^{c} \ell_b(f_j'(\boldsymbol{x}_i), y_{ij}) \right)^2}$$

$$\leq \sqrt{\frac{1}{n} \sum_{i=1}^{n} \frac{1}{c} \sum_{j=1}^{c} \left( \ell_b(f_j(\boldsymbol{x}_i), y_{ij}) - \ell_b(f_j'(\boldsymbol{x}_i), y_{ij}) \right)^2} \quad \text{(Use Jensen's Inequality)}$$

$$\leq \sqrt{\frac{1}{n} \sum_{i=1}^{n} \frac{1}{c} \sum_{j=1}^{c} 6\gamma(\ell_b(f_j(\boldsymbol{x}_i), y_{ij}) + \ell_b(f_j'(\boldsymbol{x}_i), y_{ij}))(f_j(\boldsymbol{x}_i) - f_j'(\boldsymbol{x}_i))^2} \quad \text{(Use Lemma A.1)}$$

$$\leq \sqrt{\frac{1}{n} \sum_{i=1}^{n} \frac{1}{c} \sum_{j=1}^{c} 6\gamma(\ell_b(f_j(\boldsymbol{x}_i), y_{ij}) + \ell_b(f_j'(\boldsymbol{x}_i), y_{ij})) \sqrt{\max_{i \in [n], j \in [c]} (f_j(\boldsymbol{x}_i) - f_j'(\boldsymbol{x}_i))^2}}$$

$$\leq \sqrt{12\gamma M} \max_{i \in [n], j \in [c]} |f_j(\boldsymbol{x}_i) - f_j'(\boldsymbol{x}_i)| \quad \text{(Use Assumption 4.1)}$$

$$\leq \sqrt{12\gamma M} \max_i \max_j |p_j(\boldsymbol{f}(\boldsymbol{x}_i)) - p_j(\boldsymbol{f}'(\boldsymbol{x}_i))|. \quad \text{(The definition of the projection function class $\mathcal{P}(\mathcal{F})$)}$$

Then, according to the definitions of the empirical $\ell_2$ and $\ell_\infty$ covering number, we have that an empirical $\ell_\infty$ cover of $\mathcal{P}(\mathcal{F})$ at radius $\epsilon/\sqrt{12\gamma M}$ is also an empirical $\ell_2$ cover of the loss function space $\mathcal{L}_H$ at radius $\epsilon$, and we can conclude that:

$$\mathcal{N}_2(\epsilon, \mathcal{L}_H, D) \leq \mathcal{N}_\infty \left( \frac{\epsilon}{\sqrt{12\gamma M}}, \mathcal{P}(\mathcal{F}), [c] \times D \right). \tag{5}$$

**Step 2**: We show that the empirical $\ell_\infty$ norm covering number of $\mathcal{P}(\mathcal{F})$ can be bounded by the fat-shattering dimension, and the fat-shattering dimension can be bounded by the worst-case Rademacher complexity of $\mathcal{P}(\mathcal{F})$.

According to Lemma A.3, for any $\epsilon > 2\hat{\Re}_{[c]\times D}(\mathcal{P}(\mathcal{F}))$, we have

$$\mathrm{fat}_\epsilon(\mathcal{P}(\mathcal{F})) \leq \frac{4nc\hat{\Re}^2_{[c]\times D}(\mathcal{P}(\mathcal{F}))}{\epsilon^2}.$$

Then, combining with Lemma A.4, for any $\epsilon \in (0, 2B]$, we have

$$\log \mathcal{N}_\infty\left(\epsilon, \mathcal{P}(\mathcal{F}), [c]\times D\right) \leq 1 + \mathrm{fat}_{\epsilon/4}(\mathcal{P}(\mathcal{F})) \log_2^2 \frac{8eB^2 nc}{\epsilon^2}$$

$$\leq 1 + \frac{64nc\hat{\Re}^2_{[c]\times D}(\mathcal{P}(\mathcal{F}))}{\epsilon^2} \log_2^2 \frac{8eB^2 nc}{\epsilon^2}$$

$$\leq 1 + \frac{64nc\widetilde{\Re}^2_{nc}(\mathcal{P}(\mathcal{F}))}{\epsilon^2} \log_2^2 \frac{8eB^2 nc}{\epsilon^2}. \tag{6}$$

**Step 3**: According to Assumption 4.1 in the main paper, we can obtain the lower bound of the worst-case Rademacher complexity $\widetilde{\Re}_{nc}(\mathcal{P}(\mathcal{F}))$ by the Khintchine-Kahane inequality with $p = 1$:

$$\widetilde{\Re}_{nc}(\mathcal{P}(\mathcal{F}))$$

$$= \sup_{[c]\times D\in[c]\times\mathcal{X}^n} \hat{\Re}_{[c]\times D}(\mathcal{P}(\mathcal{F}))$$

$$= \sup_{[c]\times D\in[c]\times\mathcal{X}^n} \mathbb{E}_{\boldsymbol{\epsilon}}\left[\sup_{p_j(\boldsymbol{f}(\boldsymbol{x}_i))\in\mathcal{P}(\mathcal{F})} \frac{1}{nc}\sum_{i=1}^n\sum_{j=1}^c \epsilon_{ij}p_j(\boldsymbol{f}(\boldsymbol{x}_i))\right]$$

$$= \sup_{[c]\times D\in[c]\times\mathcal{X}^n} \mathbb{E}_{\boldsymbol{\epsilon}}\left[\sup_{f_j\in\mathcal{F}_j} \frac{1}{nc}\sum_{i=1}^n\sum_{j=1}^c \epsilon_{ij}f_j(\boldsymbol{x}_i)\right]$$

$$= \sup_{\|\phi_j(\boldsymbol{x}_i)\|\leq A:i\in[n],j\in[c]} \frac{1}{nc}\mathbb{E}_{\boldsymbol{\epsilon}}\left[\sup_{\|\boldsymbol{w}_j\|\leq\Lambda} \sum_{i=1}^n\sum_{j=1}^c \epsilon_{ij}\langle\boldsymbol{w}_j, \phi_j(\boldsymbol{x}_i)\rangle\right]$$

$$= \sup_{\|\phi_j(\boldsymbol{x}_i)\|\leq A:i\in[n],j\in[c]} \frac{\Lambda}{nc}\mathbb{E}_{\boldsymbol{\epsilon}}\|\sum_{i=1}^n\sum_{j=1}^c \epsilon_{ij}\phi_j(\boldsymbol{x}_i)\|$$

$$\geq \sup_{\|\phi_j(\boldsymbol{x}_i)\|\leq A:i\in[n],j\in[c]} \frac{\Lambda}{nc}\frac{1}{\sqrt{2}}\left[\sum_{i=1}^n\sum_{j=1}^c \|\phi_j(\boldsymbol{x}_i)\|^2\right]^{\frac{1}{2}}. \quad \text{(Use Lemma A.2)}$$

Since $\|\phi_j(\boldsymbol{x}_i)\| \leq A$, we set $\sup_{\|\phi_j(\boldsymbol{x}_i)\|\leq A:i\in[n],j\in[c]} \frac{1}{nc}\left[\sum_{i=1}^n\sum_{j=1}^c \|\phi_j(\boldsymbol{x}_i)\|^2\right]^{\frac{1}{2}} = \frac{A}{\sqrt{nc}}$. So,

$$\widetilde{\Re}_{nc}(\mathcal{P}(\mathcal{F})) \geq \frac{\Lambda A}{\sqrt{2nc}} := \frac{B}{\sqrt{2nc}}. \tag{7}$$

**Step 4**: According to Lemma A.5 and combined with the above steps, we have

$$\hat{\Re}_D(\mathcal{L}_H)$$

$$\leq \inf_{\alpha>0}\left(4\alpha + \frac{12}{\sqrt{n}}\int_\alpha^M \sqrt{\log\mathcal{N}_2(\epsilon, \mathcal{L}, D)}d\epsilon\right)$$

$$\leq \inf_{\alpha>0}\left(4\alpha + \frac{12}{\sqrt{n}}\int_\alpha^M \sqrt{\log\mathcal{N}_\infty\left(\frac{\epsilon}{\sqrt{12\gamma M}}, \mathcal{P}(\mathcal{F}), [c]\times D\right)}d\epsilon\right) \quad \text{(Use inequality (5))}$$

$$\leq \inf_{\alpha>0} \left( 4\alpha + \frac{12}{\sqrt{n}} \int_\alpha^M \sqrt{1 + \frac{64nc12\gamma M \widetilde{\mathfrak{R}}_{nc}^2(\mathcal{P}(\mathcal{F}))}{\epsilon^2} \log_2^2 \frac{24eB^2 nc\gamma M}{\widetilde{\mathfrak{R}}_{nc}^2(\mathcal{P}(\mathcal{F}))}} d\epsilon \right) \quad \text{(Use inequality (6))}$$

$$\leq \inf_{\alpha>0} \left( 4\alpha + \frac{12}{\sqrt{n}} \int_\alpha^M \sqrt{1 + \frac{64nc12\gamma M \widetilde{\mathfrak{R}}_{nc}^2(\mathcal{P}(\mathcal{F}))}{\epsilon^2} \log_2^2(48en^2c^2\gamma M)} d\epsilon \right) \quad \text{(Use inequality (7))}$$

$$\leq \inf_{\alpha>0} \left( 4\alpha + \frac{12M}{\sqrt{n}} + 192\sqrt{3c\gamma M}\widetilde{\mathfrak{R}}_{nc}(\mathcal{P}(\mathcal{F})) \log_2(48en^2c^2\gamma M) \int_\alpha^M \epsilon^{-1} d\epsilon \right)$$

$$\leq \frac{12M}{\sqrt{n}} + \inf_{\alpha>0} \left( 4\alpha + 192\sqrt{3c\gamma M}\widetilde{\mathfrak{R}}_{nc}(\mathcal{P}(\mathcal{F})) \log_2(48en^2c^2\gamma M) \cdot \ln \frac{M}{\alpha} \right)$$

$$\leq \frac{12M}{\sqrt{n}} + 192\sqrt{3c\gamma M}\widetilde{\mathfrak{R}}_{nc}(\mathcal{P}(\mathcal{F}))(1 + \log_2(48en^2c^2\gamma M) \cdot \ln \frac{M}{48\sqrt{3c\gamma M}\widetilde{\mathfrak{R}}_{nc}(\mathcal{P}(\mathcal{F}))})$$

$$\text{(Choose } \alpha = 48\sqrt{3c\gamma M}\widetilde{\mathfrak{R}}_{nc}(\mathcal{P}(\mathcal{F})))$$

$$\leq \frac{12M}{\sqrt{n}} + 192\sqrt{3c\gamma M}\widetilde{\mathfrak{R}}_{nc}(\mathcal{P}(\mathcal{F}))(1 + \log_2(48en^2c^2\gamma M) \cdot \ln \frac{\sqrt{nM}}{\sqrt{\gamma}B}). \quad \text{(Use inequality (7))}$$

Second, we derive the tight vector-contraction inequality for Subset loss with smooth base loss functions.

**Step 1**: We first derive the relationship between the empirical $\ell_2$ norm covering number $\mathcal{N}_2(\epsilon, \mathcal{L}_S, D)$ and the empirical $\ell_\infty$ norm covering number $\mathcal{N}_\infty(\epsilon, \mathcal{P}(\mathcal{F}), [c] \times D)$ by the smoothness of base loss functions.

For the dataset $D = \{(\boldsymbol{x}_1, \boldsymbol{y}_1), \ldots, (\boldsymbol{x}_n, \boldsymbol{y}_n)\}$ with $n$ i.i.d. examples:

$$\sqrt{\frac{1}{n} \sum_{i=1}^n \left( \ell_S(\boldsymbol{f}(\boldsymbol{x}_i), \boldsymbol{y}_i) - \ell_S(\boldsymbol{f}'(\boldsymbol{x}_i), \boldsymbol{y}_i) \right)^2}$$

$$= \sqrt{\frac{1}{n} \sum_{i=1}^n \left( \max_{j \in [c]} \{ \ell_b(f_j(\boldsymbol{x}_i), y_{ij}) \} - \max_{j \in [c]} \{ \ell_b(f'_j(\boldsymbol{x}_i), y_{ij}) \} \right)^2}$$

$$\leq \sqrt{\frac{1}{n} \sum_{i=1}^n \max_{j \in [c]} \left( \ell_b(f_j(\boldsymbol{x}_i), y_{ij}) - \ell_b(f'_j(\boldsymbol{x}_i), y_{ij}) \right)^2}$$

$$\leq \sqrt{\frac{1}{n} \sum_{i=1}^n \max_{j \in [c]} 6\gamma(\ell_b(f_j(\boldsymbol{x}_i), y_{ij}) + \ell_b(f'_j(\boldsymbol{x}_i), y_{ij}))(f_j(\boldsymbol{x}_i) - f'_j(\boldsymbol{x}_i))^2} \quad \text{(Use Lemma A.1)}$$

$$\leq \sqrt{\frac{6\gamma}{n} \sum_{i=1}^n \max_{j \in [c]}(\ell_b(f_j(\boldsymbol{x}_i), y_{ij}) + \ell_b(f'_j(\boldsymbol{x}_i), y_{ij})) \sqrt{\max_{i \in [n], j \in [c]}(f_j(\boldsymbol{x}_i) - f'_j(\boldsymbol{x}_i))^2}}$$

$$\leq \sqrt{12\gamma M} \max_{i \in [n], j \in [c]} |f_j(\boldsymbol{x}_i) - f'_j(\boldsymbol{x}_i)| \quad \text{(Use Assumption 4.1)}$$

$$\leq \sqrt{12\gamma M} \max_i \max_j |p_j(\boldsymbol{f}(\boldsymbol{x}_i)) - p_j(\boldsymbol{f}'(\boldsymbol{x}_i))|. \quad \text{(The definition of the projection function class } \mathcal{P}(\mathcal{F}))$$

Then, according to the definitions of the empirical $\ell_2$ and $\ell_\infty$ covering number, we have that an empirical $\ell_\infty$ cover of $\mathcal{P}(\mathcal{F})$ at radius $\epsilon/\sqrt{12\gamma M}$ is also an empirical $\ell_2$ cover of the loss function space $\mathcal{L}_S$ at radius $\epsilon$, and we can conclude that:

$$\mathcal{N}_2(\epsilon, \mathcal{L}_S, D) \leq \mathcal{N}_\infty \left( \frac{\epsilon}{\sqrt{12\gamma M}}, \mathcal{P}(\mathcal{F}), [c] \times D \right).$$

**Step 2**: The following proof process is similar to the proof **Step 2-4** of the tight vector-contraction inequality for Hamming

loss. Hence, we have:

$$\hat{\mathfrak{R}}_D(\mathcal{L}_S) \leq \frac{12M}{\sqrt{n}} + 192\sqrt{3c\gamma M}\widetilde{\mathfrak{R}}_{nc}(\mathcal{P}(\mathcal{F}))(1 + \log_2(48en^2c^2\gamma M) \cdot \ln\frac{\sqrt{nM}}{\sqrt{\gamma}B}).$$

Third, we derive the tight vector-contraction inequality for Ranking loss with smooth base loss functions.

**Step 1**: We first derive the relationship between the empirical $\ell_2$ norm covering number $\mathcal{N}_2(\epsilon, \mathcal{L}_R, D)$ and the empirical $\ell_\infty$ norm covering number $\mathcal{N}_\infty(\epsilon, \mathcal{P}(\mathcal{F}), [c] \times D)$ by the smoothness of base loss functions.

For the dataset $D = \{(\boldsymbol{x}_1, \boldsymbol{y}_1), \ldots, (\boldsymbol{x}_n, \boldsymbol{y}_n)\}$ with $n$ i.i.d. examples:

$$\sqrt{\frac{1}{n}\sum_{i=1}^n \left(\ell_R(\boldsymbol{f}(\boldsymbol{x}_i), \boldsymbol{y}_i) - \ell_R(\boldsymbol{f}'(\boldsymbol{x}_i), \boldsymbol{y}_i)\right)^2}$$

$$= \sqrt{\frac{1}{n}\sum_{i=1}^n \left(\frac{1}{|Y^+||Y^-|}\sum_{p\in Y^+}\sum_{q\in Y^-} \ell_b(f_p(\boldsymbol{x}_i) - f_q(\boldsymbol{x}_i)) - \frac{1}{|Y^+||Y^-|}\sum_{p\in Y^+}\sum_{q\in Y^-} \ell_b\left(f'_p(\boldsymbol{x}_i) - f'_q(\boldsymbol{x}_i)\right)\right)^2}$$

$$\leq \sqrt{\frac{1}{n}\sum_{i=1}^n \frac{1}{|Y^+||Y^-|}\sum_{p\in Y^+}\sum_{q\in Y^-} \left(\ell_b(f_p(\boldsymbol{x}_i) - f_q(\boldsymbol{x}_i)) - \ell_b\left(f'_p(\boldsymbol{x}_i) - f'_q(\boldsymbol{x}_i)\right)\right)^2} \quad \text{(Use Jensen's Inequality)}$$

$$\leq \sqrt{\frac{1}{n}\sum_{i=1}^n \frac{1}{|Y^+||Y^-|}\sum_{p\in Y^+}\sum_{q\in Y^-} 6\gamma(\ell_b(f_p(\boldsymbol{x}_i) - f_q(\boldsymbol{x}_i)) + \ell_b\left(f'_p(\boldsymbol{x}_i) - f'_q(\boldsymbol{x}_i)\right))(f_p(\boldsymbol{x}_i) - f_q(\boldsymbol{x}_i) - f'_p(\boldsymbol{x}_i) + f'_q(\boldsymbol{x}_i))^2}$$

(Use Lemma A.1)

$$\leq \left(\frac{6\gamma}{n}\sum_{i=1}^n \frac{1}{|Y^+||Y^-|}\sum_{p\in Y^+}\sum_{q\in Y^-} \left(\ell_b(f_p(\boldsymbol{x}_i) - f_q(\boldsymbol{x}_i)) + \ell_b\left(f'_p(\boldsymbol{x}_i) - f'_q(\boldsymbol{x}_i)\right)\right)\times\right.$$

$$\left.\sqrt{\max_{i\in[n],p\in Y^+,q\in Y^-}(f_p(\boldsymbol{x}_i) - f_q(\boldsymbol{x}_i) - f'_p(\boldsymbol{x}_i) + f'_q(\boldsymbol{x}_i))^2}\right.$$

$$\leq \sqrt{12\gamma M}\max_{i\in[n],p\in Y^+,q\in Y^-}|f_p(\boldsymbol{x}_i) - f_q(\boldsymbol{x}_i) - f'_p(\boldsymbol{x}_i) + f'_q(\boldsymbol{x}_i)| \quad \text{(Use Assumption 4.1)}$$

$$\leq \sqrt{12\gamma M}\max_{i\in[n],p\in Y^+,q\in Y^-}|(f_p(\boldsymbol{x}_i) - f'_p(\boldsymbol{x}_i)) - (f_q(\boldsymbol{x}_i) - f'_q(\boldsymbol{x}_i))|$$

$$\leq \sqrt{12\gamma M}2\max_{i\in[n],j\in[c]}|f_j(\boldsymbol{x}_i) - f'_j(\boldsymbol{x}_i)|$$

$$\leq 2\sqrt{12\gamma M}\max_i\max_j|p_j(\boldsymbol{f}(\boldsymbol{x}_i)) - p_j(\boldsymbol{f}'(\boldsymbol{x}_i))|. \quad \text{(The definition of the projection function class } \mathcal{P}(\mathcal{F}))$$

Then, according to the definitions of the empirical $\ell_2$ and $\ell_\infty$ covering number, we have that an empirical $\ell_\infty$ cover of $\mathcal{P}(\mathcal{F})$ at radius $\epsilon/2\sqrt{12\gamma M}$ is also an empirical $\ell_2$ cover of the loss function space $\mathcal{L}_R$ at radius $\epsilon$, and we can conclude that:

$$\mathcal{N}_2(\epsilon, \mathcal{L}_R, D) \leq \mathcal{N}_\infty\left(\frac{\epsilon}{2\sqrt{12\gamma M}}, \mathcal{P}(\mathcal{F}), [c] \times D\right).$$

**Step 2**: The following proof process is similar to the proof **Step 2-4** of the tight vector-contraction inequality for Hamming loss. Hence, we have:

$$\hat{\mathfrak{R}}_D(\mathcal{L}_R) \leq \frac{12M}{\sqrt{n}} + 384\sqrt{3c\gamma M}\widetilde{\mathfrak{R}}_{nc}(\mathcal{P}(\mathcal{F}))\left(1 + \log_2(192en^2c^2\gamma M) \cdot \ln\frac{\sqrt{nM}}{\sqrt{\gamma}B}\right).$$

A.3.2. PROOF OF THEOREM 4.4

We upper bound the worst-case Rademacher complexity $\widetilde{\mathfrak{R}}_{nc}(\mathcal{P}(\mathcal{F}))$ as the following:

$$
\begin{aligned}
&\widetilde{\mathfrak{R}}_{nc}(\mathcal{P}(\mathcal{F}))\\
&= \sup_{[c]\times D\in[c]\times\mathcal{X}^n} \hat{\mathfrak{R}}_{[c]\times D}(\mathcal{P}(\mathcal{F}))\\
&= \sup_{[c]\times D\in[c]\times\mathcal{X}^n} \mathbb{E}_{\boldsymbol{\epsilon}}\left[\sup_{p_j(\boldsymbol{f}(\boldsymbol{x}_i))\in\mathcal{P}(\mathcal{F})} \frac{1}{nc}\sum_{i=1}^{n}\sum_{j=1}^{c}\epsilon_{ij}p_j(\boldsymbol{f}(\boldsymbol{x}_i))\right]\\
&= \sup_{[c]\times D\in[c]\times\mathcal{X}^n} \mathbb{E}_{\boldsymbol{\epsilon}}\left[\sup_{f_j\in\mathcal{F}_j} \frac{1}{nc}\sum_{i=1}^{n}\sum_{j=1}^{c}\epsilon_{ij}f_j(\boldsymbol{x}_i)\right]\\
&= \sup_{\|\phi_j(\boldsymbol{x}_i)\|\leq A:i\in[n],j\in[c]} \frac{1}{nc}\mathbb{E}_{\boldsymbol{\epsilon}}\left[\sup_{\|\boldsymbol{w}_j\|\leq\Lambda} \sum_{i=1}^{n}\sum_{j=1}^{c}\epsilon_{ij}\langle\boldsymbol{w}_j,\phi_j(\boldsymbol{x}_i)\rangle\right]\\
&= \sup_{\|\phi_j(\boldsymbol{x}_i)\|\leq A:i\in[n],j\in[c]} \frac{\Lambda}{nc}\mathbb{E}_{\boldsymbol{\epsilon}}\|\sum_{i=1}^{n}\sum_{j=1}^{c}\epsilon_{ij}\phi_j(\boldsymbol{x}_i)\|\\
&\leq \sup_{\|\phi_j(\boldsymbol{x}_i)\|\leq A:i\in[n],j\in[c]} \frac{\Lambda}{nc}\left[\mathbb{E}_{\boldsymbol{\epsilon}}\|\sum_{i=1}^{n}\sum_{j=1}^{c}\epsilon_{ij}\phi_j(\boldsymbol{x}_i)\|^2\right]^{\frac{1}{2}} \quad\text{(Use Jensen's Inequality)}\\
&\leq \sup_{\|\phi_j(\boldsymbol{x}_i)\|\leq A:i\in[n],j\in[c]} \frac{\Lambda}{nc}\left[\sum_{i=1}^{n}\sum_{j=1}^{c}\|\phi_j(\boldsymbol{x}_i)\|^2\right]^{\frac{1}{2}} \leq \frac{\Lambda A}{\sqrt{nc}} := \frac{B}{\sqrt{nc}}. \quad\text{(Use Lemma A.2)} \quad\quad (8)
\end{aligned}
$$

Then, we have

$$
\begin{aligned}
\hat{\mathfrak{R}}_D(\mathcal{L}_H) \text{ and } \hat{\mathfrak{R}}_D(\mathcal{L}_S) &\leq \frac{12M}{\sqrt{n}} + 192\sqrt{3c\gamma M}\widetilde{\mathfrak{R}}_{nc}(\mathcal{P}(\mathcal{F}))\left(1 + \log_2(48en^2c^2\gamma M)\cdot\ln\frac{\sqrt{nM}}{\sqrt{\gamma}B}\right)\\
&\leq \frac{12M}{\sqrt{n}} + \frac{192\sqrt{3\gamma M}B\left(1 + \log_2(48en^2c^2\gamma M)\cdot\ln\frac{\sqrt{nM}}{\sqrt{\gamma}B}\right)}{\sqrt{n}},
\end{aligned}
$$

and

$$
\begin{aligned}
\hat{\mathfrak{R}}_D(\mathcal{L}_R) &\leq \frac{12M}{\sqrt{n}} + 384\sqrt{3c\gamma M}\widetilde{\mathfrak{R}}_{nc}(\mathcal{P}(\mathcal{F}))\left(1 + \log_2(192en^2c^2\gamma M)\cdot\ln\frac{\sqrt{nM}}{\sqrt{\gamma}B}\right)\\
&\leq \frac{12M}{\sqrt{n}} + \frac{384\sqrt{3\gamma M}B\left(1 + \log_2(192en^2c^2\gamma M)\cdot\ln\frac{\sqrt{nM}}{\sqrt{\gamma}B}\right)}{\sqrt{n}}.
\end{aligned}
$$

Combining with (4), then for Hamming and Subset loss with smooth base losses, we have

$$
R(\boldsymbol{f}) \leq \widehat{R}_D(\boldsymbol{f}) + \frac{24M}{\sqrt{n}} + \frac{384\sqrt{3\gamma M}B\left(1 + \log_2(48en^2c^2\gamma M)\cdot\ln\frac{\sqrt{nM}}{\sqrt{\gamma}B}\right)}{\sqrt{n}} + 3M\sqrt{\frac{\log\frac{2}{\delta}}{2n}},
$$

for Ranking loss with smooth base losses, we have

$$
R(\boldsymbol{f}) \leq \widehat{R}_D(\boldsymbol{f}) + \frac{24M}{\sqrt{n}} + \frac{768\sqrt{3\gamma M}B\left(1 + \log_2(192en^2c^2\gamma M)\cdot\ln\frac{\sqrt{nM}}{\sqrt{\gamma}B}\right)}{\sqrt{n}} + 3M\sqrt{\frac{\log\frac{2}{\delta}}{2n}}.
$$

## A.4. Faster Bounds for Smooth Base Losses

### A.4.1. PROOF OF LEMMA 5.2

**Proof Sketch**: For Hamming loss, according to the smoothness of base loss functions, we first derive the relationship between the empirical $\ell_2$ norm covering number $\mathcal{N}_2(\epsilon, \mathcal{L}_H^r, D)$ of the local Hamming loss space $\mathcal{L}_H^r$ and the empirical $\ell_\infty$ norm covering number $\mathcal{N}_\infty(\epsilon, \mathcal{P}(\mathcal{F}), [c] \times D)$ of the projection function class. Then, since the empirical $\ell_\infty$ norm covering number $\mathcal{N}_\infty(\epsilon, \mathcal{P}(\mathcal{F}), [c] \times D)$ can be bounded by the worst-case Rademacher complexity of the projection function class $\mathcal{P}(\mathcal{F})$, combining the above results and the discretized variant of Dudley's integral inequality, the local Rademacher complexity of the local Hamming loss function space $\mathcal{L}_H^r$ can be bounded by the worst-case Rademacher complexity of the projection function class, and the desired bound can be derived. For Subset loss and Ranking loss, we derive the relationships between the empirical $\ell_2$ norm covering number $\mathcal{N}_2(\epsilon, \mathcal{L}_S^r, D)$, $\mathcal{N}_2(\epsilon, \mathcal{L}_R^r, D)$ and the empirical $\ell_\infty$ norm covering number $\mathcal{N}_\infty(\epsilon, \mathcal{P}(\mathcal{F}), [c] \times D)$, respectively. Then, using the similar technique to the local vector-contraction inequality for Hamming loss, the desired bounds for Subset and Ranking loss can be derived.

We first introduce the following lemmas:

**Lemma A.6** (Discretized variant of Dudley's integral inequality in (Guermeur, 2017)). *Let $\mathcal{F}$ be a real-valued function class and $S := \{s_1, \ldots, s_n\}$ be a finite sample of size $n$. Let $(\epsilon_j)_{j=0}^\infty$ be a monotone sequence decreasing to $0$ and any $(a_1, \ldots, a_n) \in \mathbb{R}^n$. If*

$$\epsilon_0 \geq \sqrt{n^{-1} \sup_{f \in \mathcal{F}} \sum_{i=1}^n (f(s_i) - a_i)^2},$$

*then for any non-negative integer $N$ we have the following relationship between the Rademacher complexity $\hat{\Re}_S(\mathcal{F})$ and the covering number $\mathcal{N}_2(\epsilon_j, \mathcal{F}, S)$*

$$\hat{\Re}_S(\mathcal{F}) \leq 2 \sum_{j=1}^N (\epsilon_j + \epsilon_{j-1}) \sqrt{\frac{\log \mathcal{N}_2(\epsilon_j, \mathcal{F}, S)}{n}} + \epsilon_N.$$

**Lemma A.7** ((Pollard, 2012)). *Let $\mathcal{F}$ be a class of functions from $\mathcal{X}$ to $\mathbb{R}$ and let $\mathcal{F}_0$ be a subset. Then for any $\varepsilon > 0$, we have the following relationship on covering numbers: $\mathcal{N}_p(\epsilon, \mathcal{F}_0, D) \leq \mathcal{N}_p(\epsilon/2, \mathcal{F}, D)$.*

First, we derive the local vector-contraction inequality for Hamming loss with smooth base loss functions.

**Step 1**: We first derive the relationship between the empirical $\ell_2$ norm covering number $\mathcal{N}_2(\epsilon, \mathcal{L}_H^r, D)$ and the empirical $\ell_\infty$ norm covering number $\mathcal{N}_\infty(\epsilon, \mathcal{P}(\mathcal{F}), [c] \times D)$ by the smoothness of base loss functions.

For the dataset $D = \{(x_1, y_1), \ldots, (x_n, y_n)\}$ with $n$ i.i.d. examples:

$$\sqrt{\frac{1}{n} \sum_{i=1}^n \left(\ell_H(f(x_i), y_i) - \ell_H(f'(x_i), y_i)\right)^2}$$

$$= \sqrt{\frac{1}{n} \sum_{i=1}^n \left(\frac{1}{c} \sum_{j=1}^c \ell_b(f_j(x_i), y_{ij}) - \frac{1}{c} \sum_{j=1}^c \ell_b(f_j'(x_i), y_{ij})\right)^2}$$

$$\leq \sqrt{\frac{1}{n} \sum_{i=1}^n \frac{1}{c} \sum_{j=1}^c \left(\ell_b(f_j(x_i), y_{ij}) - \ell_b(f_j'(x_i), y_{ij})\right)^2} \quad \text{(Use Jensen's Inequality)}$$

$$\leq \sqrt{\frac{1}{n} \sum_{i=1}^n \frac{1}{c} \sum_{j=1}^c 6\gamma(\ell_b(f_j(x_i), y_{ij}) + \ell_b(f_j'(x_i), y_{ij}))(f_j(x_i) - f_j'(x_i))^2} \quad \text{(Use Lemma A.1)}$$

$$\leq \sqrt{\frac{1}{n} \sum_{i=1}^n \frac{1}{c} \sum_{j=1}^c 6\gamma(\ell_b(f_j(x_i), y_{ij}) + \ell_b(f_j'(x_i), y_{ij}))} \sqrt{\max_{i \in [n], j \in [c]}(f_j(x_i) - f_j'(x_i))^2}$$

$$\leq \sqrt{12\gamma r} \max_{i \in [n], j \in [c]} |f_j(x_i) - f_j'(x_i)| \quad \text{(Use Definition 5.1)}$$

$$\leq \sqrt{12\gamma r} \max_i \max_j |p_j(\boldsymbol{f}(\boldsymbol{x}_i)) - p_j(\boldsymbol{f}'(\boldsymbol{x}_i))|. \quad \text{(The definition of the projection function class } \mathcal{P}(\mathcal{F}))$$

Then, according to the definitions of the empirical $\ell_2$ and $\ell_\infty$ covering number, we have that an empirical $\ell_\infty$ cover of $\mathcal{P}(\mathcal{F}^r)$ at radius $\epsilon/\sqrt{12\gamma r}$ is also an empirical $\ell_2$ cover of the local loss function space $\mathcal{L}_H^r$ at radius $\epsilon$, and combined with Lemma A.7, we can conclude that:

$$\mathcal{N}_2\left(\epsilon, \mathcal{L}_H^r, D\right) \leq \mathcal{N}_\infty\left(\frac{\epsilon}{\sqrt{12\gamma r}}, \mathcal{P}(\mathcal{F}^r), [c] \times D\right) \leq \mathcal{N}_\infty\left(\frac{\epsilon}{2\sqrt{12\gamma r}}, \mathcal{P}(\mathcal{F}), [c] \times D\right). \tag{9}$$

**Step 2**: According to the inequality (6), we have that the empirical $\ell_\infty$ norm covering number of $\mathcal{P}(\mathcal{F})$ can be bounded by the worst-case Rademacher complexity of $\mathcal{P}(\mathcal{F})$. Let $\epsilon_N = 48\sqrt{3\gamma r} \max\{\frac{1}{\sqrt{n}}, \sqrt{c}\widetilde{\mathfrak{R}}_{nc}(\mathcal{P}(\mathcal{F}))\}$ and $\epsilon_j = 2^{N-j}\epsilon_N$, where $N = \lceil \log_2 \frac{4\sqrt{3\gamma r}B}{48\sqrt{3\gamma r}\max\{\frac{1}{\sqrt{n}}, \sqrt{c}\widetilde{\mathfrak{R}}_{nc}(\mathcal{P}(\mathcal{F}))\}} \rceil$. According to Lemma A.6 and combined with the above steps, we have

$$\hat{\mathfrak{R}}_D(\mathcal{L}_H^r)$$

$$\leq 2\sum_{j=1}^{N} (\epsilon_j + \epsilon_{j-1}) \sqrt{\frac{\log \mathcal{N}_\infty\left(\epsilon_j, \mathcal{L}_H^r, D\right)}{n}} + \epsilon_N$$

$$\leq 2\sum_{j=1}^{N} (\epsilon_j + \epsilon_{j-1}) \sqrt{\frac{\log \mathcal{N}_\infty\left(\frac{\epsilon_j}{2\sqrt{12\gamma r}}, \mathcal{P}(\mathcal{F}), [c] \times D\right)}{n}} + \epsilon_N \quad \text{(Use inequality (9))}$$

$$\leq 2\sum_{j=1}^{N} (\epsilon_j + \epsilon_{j-1}) \frac{1}{\sqrt{n}} + 2\sum_{j=1}^{N} (\epsilon_j + \epsilon_{j-1}) \sqrt{\frac{\frac{64nc(2\sqrt{12\gamma r})^2 \widetilde{\mathfrak{R}}_{nc}^2(\mathcal{P}(\mathcal{F}))}{\epsilon_j^2} \log_2^2 \frac{8eB^2nc(2\sqrt{12\gamma r})^2}{\epsilon_j^2}}{n}} + \epsilon_N$$

(Use inequality (6))

$$\leq \frac{6\epsilon_0}{\sqrt{n}} + 2\sum_{j=1}^{N} (\epsilon_j + \epsilon_{j-1}) \sqrt{\frac{\frac{64nc(2\sqrt{12\gamma r})^2 \widetilde{\mathfrak{R}}_{nc}^2(\mathcal{P}(\mathcal{F}))}{\epsilon_j^2} \log_2^2 \frac{8eB^2nc(2\sqrt{12\gamma r})^2}{\epsilon_j^2}}{n}} + \epsilon_N$$

$$\leq \frac{48\sqrt{3\gamma r}B}{\sqrt{n}} + 2\sum_{j=1}^{N} (\epsilon_j + \epsilon_{j-1}) \frac{8\sqrt{c}2\sqrt{12\gamma r}\widetilde{\mathfrak{R}}_{nc}(\mathcal{P}(\mathcal{F}))}{\epsilon_j} \log_2 \frac{8eB^2nc(2\sqrt{12\gamma r})^2}{\epsilon_j^2} + \epsilon_N \quad \text{(Use } \frac{\epsilon_0}{2\sqrt{12\gamma r}} \leq 2B\text{)}$$

$$\leq \frac{48\sqrt{3\gamma r}B}{\sqrt{n}} + 48 \cdot 4\sqrt{3c\gamma r}\widetilde{\mathfrak{R}}_{nc}(\mathcal{P}(\mathcal{F})) \sum_{j=1}^{N} \log_2 \frac{8eB^2nc(2\sqrt{12\gamma r})^2}{\epsilon_j^2} + \epsilon_N \quad \text{(Use } \epsilon_{j-1} = 2\epsilon_j\text{)}$$

$$\leq \frac{48\sqrt{3\gamma r}B}{\sqrt{n}} + 48 \cdot 4\sqrt{3c\gamma r}\widetilde{\mathfrak{R}}_{nc}(\mathcal{P}(\mathcal{F})) \sum_{j=1}^{N} \left(\log_2 32 \cdot 12eB^2nc\gamma r + \log_2 \frac{1}{\epsilon_j^2}\right) + \epsilon_N$$

$$\leq \frac{48\sqrt{3\gamma r}B}{\sqrt{n}} + 48 \cdot 4\sqrt{3c\gamma r}\widetilde{\mathfrak{R}}_{nc}(\mathcal{P}(\mathcal{F}))N \log_2 32 \cdot 12eB^2nc\gamma r + 48 \cdot 4\sqrt{3c\gamma r}\widetilde{\mathfrak{R}}_{nc}(\mathcal{P}(\mathcal{F})) \sum_{j=1}^{N} \log_2 \frac{1}{\epsilon_j^2} + \epsilon_N$$

$$\leq \frac{48\sqrt{3\gamma r}B}{\sqrt{n}} + 48 \cdot 4\sqrt{3c\gamma r}\widetilde{\mathfrak{R}}_{nc}(\mathcal{P}(\mathcal{F}))N \log_2 32 \cdot 12eB^2nc\gamma r + 48 \cdot 4\sqrt{3c\gamma r}\widetilde{\mathfrak{R}}_{nc}(\mathcal{P}(\mathcal{F})) \sum_{j=1}^{N} \log_2 \frac{2^{2j}}{\epsilon_0^2} + \epsilon_N$$

(Use $\epsilon_j = 2^{-j} \cdot \epsilon_0$)

$$\leq \frac{48\sqrt{3\gamma r}B}{\sqrt{n}} + 48 \cdot 4\sqrt{3c\gamma r}\widetilde{\mathfrak{R}}_{nc}(\mathcal{P}(\mathcal{F}))N \log_2 32 \cdot 12eB^2nc\gamma r +$$

$$48 \cdot 4\sqrt{3c\gamma r}\widetilde{\mathfrak{R}}_{nc}(\mathcal{P}(\mathcal{F}))(\sum_{j=1}^{N} \log_2 \frac{1}{\epsilon_0^2} + 2\log_2 2 \sum_{j=1}^{N} j) + \epsilon_N$$

$$\leq \frac{48\sqrt{3\gamma r}B}{\sqrt{n}} + 48 \cdot 4\sqrt{3c\gamma r}\widetilde{\Re}_{nc}(\mathcal{P}(\mathcal{F}))N\log_2 32 \cdot 12eB^2nc\gamma r + 48 \cdot 4\sqrt{3c\gamma r}\widetilde{\Re}_{nc}(\mathcal{P}(\mathcal{F}))N(\log_2 \frac{1}{\epsilon_0^2} + (N+1)\log_2 2) + \epsilon_N$$

$$\leq \frac{48\sqrt{3\gamma r}B}{\sqrt{n}} + 48 \cdot 4\sqrt{3c\gamma r}\widetilde{\Re}_{nc}(\mathcal{P}(\mathcal{F}))N\log_2 32 \cdot 12eB^2nc\gamma r + 48 \cdot 4\sqrt{3c\gamma r}\widetilde{\Re}_{nc}(\mathcal{P}(\mathcal{F}))N\log_2(\frac{2}{\epsilon_0} \cdot \frac{1}{\epsilon_N}) + \epsilon_N$$

(Use $\epsilon_0 = 2^N \cdot \epsilon_N$)

$$\leq \frac{48\sqrt{3\gamma r}B}{\sqrt{n}} + 48 \cdot 4\sqrt{3c\gamma r}\widetilde{\Re}_{nc}(\mathcal{P}(\mathcal{F}))N\log_2 32 \cdot 12eB^2nc\gamma r +$$

$$48 \cdot 4\sqrt{3c\gamma r}\widetilde{\Re}_{nc}(\mathcal{P}(\mathcal{F}))N\log_2(\frac{2}{4\sqrt{3\gamma r}B} \cdot \frac{1}{48\sqrt{3\gamma r}\frac{1}{\sqrt{n}}}) + \epsilon_N \quad \text{(Use } \frac{\epsilon_0}{2\sqrt{12\gamma r}} \geq B \text{ and } \epsilon_N \geq 48\sqrt{3\gamma r}\frac{1}{\sqrt{n}})$$

$$\leq \frac{48\sqrt{3\gamma r}B}{\sqrt{n}} + 48 \cdot 4\sqrt{3c\gamma r}\widetilde{\Re}_{nc}(\mathcal{P}(\mathcal{F}))N\log_2 32 \cdot 12eB^2nc\gamma r + 48 \cdot 4\sqrt{3c\gamma r}\widetilde{\Re}_{nc}(\mathcal{P}(\mathcal{F}))N\log_2 \frac{\sqrt{n}}{6 \cdot 48\gamma rB} + \epsilon_N$$

$$= \frac{48\sqrt{3\gamma r}B}{\sqrt{n}} + 48 \cdot 4\sqrt{3c\gamma r}\widetilde{\Re}_{nc}(\mathcal{P}(\mathcal{F}))N(\log_2 32 \cdot 12eB^2nc\gamma r + \log_2 \frac{\sqrt{n}}{6 \cdot 48\gamma rB}) + \epsilon_N$$

$$\leq \frac{48\sqrt{3\gamma r}B}{\sqrt{n}} + 48 \cdot 4\sqrt{3c\gamma r}\widetilde{\Re}_{nc}(\mathcal{P}(\mathcal{F}))\log_2(\frac{4eBn^{\frac{3}{2}}c}{3})\lceil\log_2 \frac{4\sqrt{3\gamma r}B}{48\sqrt{3\gamma r}\frac{1}{\sqrt{n}}}\rceil + \epsilon_N$$

(Use $N = \lceil\log_2 \frac{4\sqrt{3\gamma r}B}{48\sqrt{3\gamma r}\max\{\frac{1}{\sqrt{n}}, \sqrt{c}\widetilde{\Re}_{nc}(\mathcal{P}(\mathcal{F}))\}}\rceil$)

$$\leq \frac{48\sqrt{3\gamma r}B}{\sqrt{n}} + 48 \cdot 4\sqrt{3c\gamma r}\widetilde{\Re}_{nc}(\mathcal{P}(\mathcal{F}))\log_2(\frac{4eBn^{\frac{3}{2}}c}{3})\lceil\log_2 \frac{\sqrt{n}B}{12}\rceil + \epsilon_N$$

$$\leq \frac{48\sqrt{3\gamma r}B}{\sqrt{n}} + 48 \cdot 4\sqrt{3c\gamma r}\widetilde{\Re}_{nc}(\mathcal{P}(\mathcal{F}))\log_2(4Bn^{\frac{3}{2}}c)\log_2 \frac{\sqrt{n}B}{6} + 48\sqrt{3\gamma r}\max\{\frac{1}{\sqrt{n}}, \sqrt{c}\widetilde{\Re}_{nc}(\mathcal{P}(\mathcal{F}))\}$$

$$\leq \frac{48\sqrt{3\gamma r}(B+1)}{\sqrt{n}} + 48\sqrt{3c\gamma r}\widetilde{\Re}_{nc}(\mathcal{P}(\mathcal{F}))(1 + 4\log_2(4Bn^{\frac{3}{2}}c)\log_2(\sqrt{n}B)). \quad \text{(Use } \max\{a, b\} \leq a + b)$$

Second, we derive the local vector-contraction inequality for Subset loss with smooth base loss functions.

**Step 1**: We first derive the relationship between the empirical $\ell_2$ norm covering number $\mathcal{N}_2(\epsilon, \mathcal{L}_S^r, D)$ and the empirical $\ell_\infty$ norm covering number $\mathcal{N}_\infty(\epsilon, \mathcal{P}(\mathcal{F}), [c] \times D)$ by the smoothness of base loss functions.

For the dataset $D = \{(\boldsymbol{x}_1, \boldsymbol{y}_1), \ldots, (\boldsymbol{x}_n, \boldsymbol{y}_n)\}$ with $n$ i.i.d. examples:

$$\sqrt{\frac{1}{n}\sum_{i=1}^n (\ell_S(\boldsymbol{f}(\boldsymbol{x}_i), \boldsymbol{y}_i) - \ell_S(\boldsymbol{f}'(\boldsymbol{x}_i), \boldsymbol{y}_i))^2}$$

$$= \sqrt{\frac{1}{n}\sum_{i=1}^n \left(\max_{j\in[c]}\{\ell_b(f_j(\boldsymbol{x}_i), y_{ij})\} - \max_{j\in[c]}\{\ell_b(f_j'(\boldsymbol{x}_i), y_{ij})\}\right)^2}$$

$$\leq \sqrt{\frac{1}{n}\sum_{i=1}^n \max_{j\in[c]}(\ell_b(f_j(\boldsymbol{x}_i), y_{ij}) - \ell_b(f_j'(\boldsymbol{x}_i), y_{ij}))^2}$$

$$\leq \sqrt{\frac{1}{n}\sum_{i=1}^n \max_{j\in[c]} 6\gamma(\ell_b(f_j(\boldsymbol{x}_i), y_{ij}) + \ell_b(f_j'(\boldsymbol{x}_i), y_{ij}))(f_j(\boldsymbol{x}_i) - f_j'(\boldsymbol{x}_i))^2} \quad \text{(Use Lemma A.1)}$$

$$\leq \sqrt{\frac{6\gamma}{n}\sum_{i=1}^n \max_{j\in[c]}(\ell_b(f_j(\boldsymbol{x}_i), y_{ij}) + \ell_b(f_j'(\boldsymbol{x}_i), y_{ij}))}\sqrt{\max_{i\in[n], j\in[c]}(f_j(\boldsymbol{x}_i) - f_j'(\boldsymbol{x}_i))^2}$$

$$\leq \sqrt{12\gamma r}\max_{i\in[n], j\in[c]}|f_j(\boldsymbol{x}_i) - f_j'(\boldsymbol{x}_i)| \quad \text{(Use Definition 5.1)}$$

$$\leq \sqrt{12\gamma r} \max_i \max_j |p_j(\boldsymbol{f}(\boldsymbol{x}_i)) - p_j(\boldsymbol{f}'(\boldsymbol{x}_i))|. \quad \text{(The definition of the projection function class } \mathcal{P}(\mathcal{F}))$$

Then, according to the definitions of the empirical $\ell_2$ and $\ell_\infty$ covering number, we have that an empirical $\ell_\infty$ cover of $\mathcal{P}(\mathcal{F}^r)$ at radius $\epsilon/\sqrt{12\gamma r}$ is also an empirical $\ell_2$ cover of the local loss function space $\mathcal{L}_S^r$ at radius $\epsilon$, and combined with Lemma A.7, we can conclude that:

$$\mathcal{N}_2(\epsilon, \mathcal{L}_S^r, D) \leq \mathcal{N}_\infty\left(\frac{\epsilon}{\sqrt{12\gamma r}}, \mathcal{P}(\mathcal{F}^r), [c] \times D\right) \leq \mathcal{N}_\infty\left(\frac{\epsilon}{2\sqrt{12\gamma r}}, \mathcal{P}(\mathcal{F}), [c] \times D\right).$$

**Step 2**: The following proof process is similar to the proof **Step 2** of the local vector-contraction inequality for Hamming loss. Hence, we have:

$$\hat{\mathfrak{R}}_D(\mathcal{L}_S^r) \leq \frac{48\sqrt{3\gamma r}(B+1)}{\sqrt{n}} + 48\sqrt{3c\gamma r}\widetilde{\mathfrak{R}}_{nc}(\mathcal{P}(\mathcal{F}))(1 + 4\log_2(4Bn^{\frac{3}{2}}c)\log_2(\sqrt{n}B)).$$

Third, we derive the local vector-contraction inequality for Ranking loss with smooth base loss functions.

**Step 1**: We first derive the relationship between the empirical $\ell_2$ norm covering number $\mathcal{N}_2(\epsilon, \mathcal{L}_R^r, D)$ and the empirical $\ell_\infty$ norm covering number $\mathcal{N}_\infty(\epsilon, \mathcal{P}(\mathcal{F}), [c] \times D)$ by the smoothness of base loss functions.

For the dataset $D = \{(\boldsymbol{x}_1, \boldsymbol{y}_1), \ldots, (\boldsymbol{x}_n, \boldsymbol{y}_n)\}$ with $n$ i.i.d. examples:

$$\sqrt{\frac{1}{n}\sum_{i=1}^n \left(\ell_R(\boldsymbol{f}(\boldsymbol{x}_i), \boldsymbol{y}_i) - \ell_R(\boldsymbol{f}'(\boldsymbol{x}_i), \boldsymbol{y}_i)\right)^2}$$

$$= \sqrt{\frac{1}{n}\sum_{i=1}^n \left(\frac{1}{|Y^+||Y^-|}\sum_{p\in Y^+}\sum_{q\in Y^-}\ell_b\left(f_p(\boldsymbol{x}_i) - f_q(\boldsymbol{x}_i)\right) - \frac{1}{|Y^+||Y^-|}\sum_{p\in Y^+}\sum_{q\in Y^-}\ell_b\left(f_p'(\boldsymbol{x}_i) - f_q'(\boldsymbol{x}_i)\right)\right)^2}$$

$$\leq \sqrt{\frac{1}{n}\sum_{i=1}^n \frac{1}{|Y^+||Y^-|}\sum_{p\in Y^+}\sum_{q\in Y^-}\left(\ell_b\left(f_p(\boldsymbol{x}_i) - f_q(\boldsymbol{x}_i)\right) - \ell_b\left(f_p'(\boldsymbol{x}_i) - f_q'(\boldsymbol{x}_i)\right)\right)^2} \quad \text{(Use Jensen's Inequality)}$$

$$\leq \sqrt{\frac{1}{n}\sum_{i=1}^n \frac{1}{|Y^+||Y^-|}\sum_{p\in Y^+}\sum_{q\in Y^-}6\gamma(\ell_b\left(f_p(\boldsymbol{x}_i) - f_q(\boldsymbol{x}_i)\right) + \ell_b\left(f_p'(\boldsymbol{x}_i) - f_q'(\boldsymbol{x}_i)\right))(f_p(\boldsymbol{x}_i) - f_q(\boldsymbol{x}_i) - f_p'(\boldsymbol{x}_i) + f_q'(\boldsymbol{x}_i))^2}$$

(Use Lemma A.1)

$$\leq \sqrt{\frac{6\gamma}{n}\sum_{i=1}^n \frac{1}{|Y^+||Y^-|}\sum_{p\in Y^+}\sum_{q\in Y^-}(\ell_b\left(f_p(\boldsymbol{x}_i) - f_q(\boldsymbol{x}_i)\right) + \ell_b\left(f_p'(\boldsymbol{x}_i) - f_q'(\boldsymbol{x}_i)\right))\times}$$

$$\sqrt{\max_{i\in[n], p\in Y^+, q\in Y^-}(f_p(\boldsymbol{x}_i) - f_q(\boldsymbol{x}_i) - f_p'(\boldsymbol{x}_i) + f_q'(\boldsymbol{x}_i))^2}$$

$$\leq \sqrt{12\gamma r}\max_{i\in[n], p\in Y^+, q\in Y^-}|f_p(\boldsymbol{x}_i) - f_q(\boldsymbol{x}_i) - f_p'(\boldsymbol{x}_i) + f_q'(\boldsymbol{x}_i)| \quad \text{(Use Definition 5.1)}$$

$$\leq \sqrt{12\gamma r}\max_{i\in[n], p\in Y^+, q\in Y^-}|(f_p(\boldsymbol{x}_i) - f_p'(\boldsymbol{x}_i)) - (f_q(\boldsymbol{x}_i) - f_q'(\boldsymbol{x}_i))|$$

$$\leq \sqrt{12\gamma r}2\max_{i\in[n], j\in[c]}|f_j(\boldsymbol{x}_i) - f_j'(\boldsymbol{x}_i)|$$

$$\leq 2\sqrt{12\gamma r}\max_i \max_j |p_j(\boldsymbol{f}(\boldsymbol{x}_i)) - p_j(\boldsymbol{f}'(\boldsymbol{x}_i))|. \quad \text{(The definition of the projection function class } \mathcal{P}(\mathcal{F}))$$

Then, according to the definitions of the empirical $\ell_2$ and $\ell_\infty$ covering number, we have that an empirical $\ell_\infty$ cover of $\mathcal{P}(\mathcal{F}^r)$ at radius $\epsilon/2\sqrt{12\gamma r}$ is also an empirical $\ell_2$ cover of the local loss function space $\mathcal{L}_R^r$ at radius $\epsilon$, and combined with Lemma A.7, we can conclude that:

$$\mathcal{N}_2(\epsilon, \mathcal{L}_R^r, D) \leq \mathcal{N}_\infty\left(\frac{\epsilon}{2\sqrt{12\gamma r}}, \mathcal{P}(\mathcal{F}^r), [c] \times D\right) \leq \mathcal{N}_\infty\left(\frac{\epsilon}{4\sqrt{12\gamma r}}, \mathcal{P}(\mathcal{F}), [c] \times D\right).$$

**Step 2**: The following proof process is similar to the proof **Step 2** of the local vector-contraction inequality for Hamming loss. Hence, we have:

$$\hat{\Re}_D(\mathcal{L}_S^r) \leq \frac{96\sqrt{3\gamma r}(B+1)}{\sqrt{n}} + 96\sqrt{3c\gamma r}\widetilde{\Re}_{nc}(\mathcal{P}(\mathcal{F}))(1 + 4\log_2(4Bn^{\frac{3}{2}}c)\log_2(\sqrt{n}B)).$$

A.4.2. PROOF OF THEOREM 5.5

We first introduce the following lemma:

**Lemma A.8** (Theorem 6.1 in (Bousquet, 2002)). *Let $\mathcal{F}$ be a class of non-negative functions such that $0 \leq f \leq c$ almost surely. Let $\widehat{R}_D(f) = \frac{1}{n}\sum_{i=1}^n f(\boldsymbol{x}_i)$. Let $\psi_n$ be a sub-root function such that for all $r > 0$,*

$$\hat{\Re}_D(\{f : \widehat{R}_D(f) \leq r\}) \leq \psi_n(r).$$

*Define $r_n^*$ as the largest solution of the equation $\psi_n(r) = r$. For all $x > 0$, with probability at least $1 - e^{-x}$ for all $f \in \mathcal{F}$*

$$R(f) \leq 2\widehat{R}_D(f) + 106r_n^* + 48r_0,$$

*where $r_0 = c(x + 6\log\log n)/n$.*

According to Lemma A.8, for a sub-root function $\psi_n(r)$, $r > 0$, $\hat{\Re}_D(\mathcal{L}^r) \leq \psi_n(r)$, we have the following holds with probability at least $1 - \delta$,

$$R(\boldsymbol{f}) \leq 2\widehat{R}_D(\boldsymbol{f}) + 106r_n^* + \frac{48M(\log\frac{1}{\delta} + 6\log\log n)}{n},$$

where $r_n^*$ is the largest solution of the equation $\psi_n(r) = r$, i.e., the fixed point of $\psi_n$.

For Hamming loss and Subset loss, we set $\psi_n(r) = \frac{48\sqrt{3\gamma r}(B+1)}{\sqrt{n}} + 48\sqrt{3c\gamma r}\widetilde{\Re}_{nc}(\mathcal{P}(\mathcal{F}))(1 + 4\log_2(4Bn^{\frac{3}{2}}c)\log_2(\sqrt{n}B))$, which is a sub-root function, and $\hat{\Re}_D(\mathcal{L}_H^r) \leq \psi_n(r)$, $\hat{\Re}_D(\mathcal{L}_S^r) \leq \psi_n(r)$. Hence, solving the equation $\psi_n(r) = r$ gives the fixed point

$$r_n^* = 3 \cdot 48^2\gamma\left(\frac{B+1}{\sqrt{n}} + \sqrt{c}\widetilde{\Re}_{nc}(\mathcal{P}(\mathcal{F}))(1 + 4\log_2(4Bn^{\frac{3}{2}}c)\log_2(\sqrt{n}B))\right)^2.$$

According to the inequality (8), we have that the upper bound of the worst-case Rademacher complexity is $\widetilde{\Re}_{nc}(\mathcal{P}(\mathcal{F})) \leq \frac{B}{\sqrt{nc}}$, then combined with the above steps, we have

$$R(\boldsymbol{f}) \leq 2\widehat{R}_D(\boldsymbol{f}) + \frac{3 \cdot 48^2 a\gamma(B+1)^2\left(1 + (1 + 4\log_2(4Bn^{\frac{3}{2}}c)\log_2(\sqrt{n}B))\right)^2}{n} + \frac{bM(\log\frac{1}{\delta} + 6\log\log n)}{n},$$

where $a = 106, b = 48$.

Using similar techniques and local vector-contraction inequality of Ranking loss, we can derive the sharp bound for Ranking loss with smooth base loss functions:

$$R(\boldsymbol{f}) \leq 2\widehat{R}_D(\boldsymbol{f}) + \frac{3 \cdot 96^2 a\gamma(B+1)^2\left(1 + (1 + 4\log_2(4Bn^{\frac{3}{2}}c)\log_2(\sqrt{n}B))\right)^2}{n} + \frac{bM(\log\frac{1}{\delta} + 6\log\log n)}{n},$$

where $a = 106, b = 48$.

## A.5. Tighter Bounds for Macro-Averaged AUC

A.5.1. PROOF OF LEMMA 6.2

**Proof Sketch**: We first derive the relationship between the empirical $\ell_\infty$ norm covering number $\mathcal{N}_\infty(\epsilon, \mathcal{L}, D)$ of the loss function space and the empirical $\ell_\infty$ norm covering number $\mathcal{N}_\infty(\epsilon, \mathcal{P}(\mathcal{F}), [c] \times D)$ of the projection function class. Then, since the empirical $\ell_\infty$ norm covering number $\mathcal{N}_\infty(\epsilon, \mathcal{P}(\mathcal{F}), [c] \times D)$ can be bounded by the worst-case Rademacher

complexity of the projection function class, combining the above results and the refined Dudley's integral inequality, the Rademacher complexity of the loss function space can be bounded by the worst-case Rademacher complexity of the projection function class, and the desired bound can be derived.

**Step 1**: We first derive the relationship between the empirical $\ell_\infty$ norm covering number $\mathcal{N}_\infty(\epsilon, \mathcal{L}, D)$ and the empirical $\ell_\infty$ norm covering number $\mathcal{N}_\infty(\epsilon, \mathcal{P}(\mathcal{F}), [c] \times D)$.

For the dataset $D = \{(\boldsymbol{x}_1, \boldsymbol{y}_1), \ldots, (\boldsymbol{x}_n, \boldsymbol{y}_n)\}$ with $n$ i.i.d. examples:

$$\max_i |\ell(\boldsymbol{f}(\boldsymbol{x}_i), \boldsymbol{y}_i) - \ell(\boldsymbol{f}'(\boldsymbol{x}_i), \boldsymbol{y}_i)|$$
$$\leq \rho \max_i \|\boldsymbol{f}(\boldsymbol{x}_i) - \boldsymbol{f}'(\boldsymbol{x}_i)\|_\infty \quad \text{(Use Assumption 6.1)}$$
$$\leq \rho \max_i \max_j |f_j(\boldsymbol{x}_i) - f_j'(\boldsymbol{x}_i)|$$
$$\leq \rho \max_i \max_j |p_j(\boldsymbol{f}(\boldsymbol{x}_i)) - p_j(\boldsymbol{f}'(\boldsymbol{x}_i))|. \quad \text{(The definition of the projection function class } \mathcal{P}(\mathcal{F}))$$

Then, according to the definition of the empirical $\ell_\infty$ covering number, we have that an empirical $\ell_\infty$ cover of $\mathcal{P}(\mathcal{F})$ at radius $\epsilon/\rho$ is also an empirical $\ell_\infty$ cover of the loss function space $\mathcal{L}$ at radius $\epsilon$, and we can conclude that:

$$\mathcal{N}_\infty(\epsilon, \mathcal{L}, D) \leq \mathcal{N}_\infty\left(\frac{\epsilon}{\rho}, \mathcal{P}(\mathcal{F}), [c] \times D\right). \tag{10}$$

According to the inequality (6), we have that the empirical $\ell_\infty$ norm covering number of $\mathcal{P}(\mathcal{F})$ can be bounded by the worst-case Rademacher complexity of $\mathcal{P}(\mathcal{F})$. According to Lemma A.5 and combined with the above steps, we have

$$\hat{\mathfrak{R}}_D(\mathcal{L})$$
$$\leq \inf_{\alpha>0}\left(4\alpha + \frac{12}{\sqrt{n}}\int_\alpha^M \sqrt{\log\mathcal{N}_\infty(\epsilon, \mathcal{L}, D)}d\epsilon\right)$$
$$\leq \inf_{\alpha>0}\left(4\alpha + \frac{12}{\sqrt{n}}\int_\alpha^M \sqrt{\log\mathcal{N}_\infty(\frac{\epsilon}{\rho}, \mathcal{P}(\mathcal{F}), [c] \times D)}d\epsilon\right) \quad \text{(Use inequality (10))}$$
$$\leq \inf_{\alpha>0}\left(4\alpha + \frac{12}{\sqrt{n}}\int_\alpha^M \sqrt{1 + \frac{64nc\rho^2\widetilde{\mathfrak{R}}_{nc}^2(\mathcal{P}(\mathcal{F}))}{\epsilon^2}\log_2^2\frac{2eB^2nc\rho^2}{\widetilde{\mathfrak{R}}_{nc}^2(\mathcal{P}(\mathcal{F}))}}d\epsilon\right) \quad \text{(Use inequality (6))}$$
$$\leq \inf_{\alpha>0}\left(4\alpha + \frac{12}{\sqrt{n}}\int_\alpha^M \sqrt{1 + \frac{64nc\rho^2\widetilde{\mathfrak{R}}_{nc}^2(\mathcal{P}(\mathcal{F}))}{\epsilon^2}\log_2^2(4en^2c^2\rho^2)}d\epsilon\right) \quad \text{(Use inequality (7))}$$
$$\leq \inf_{\alpha>0}\left(4\alpha + \frac{12M}{\sqrt{n}} + 96\rho\sqrt{c}\widetilde{\mathfrak{R}}_{nc}(\mathcal{P}(\mathcal{F}))\log_2(4en^2c^2\mu^2)\int_\alpha^M \epsilon^{-1}d\epsilon\right)$$
$$\leq \frac{12M}{\sqrt{n}} + \inf_{\alpha>0}\left(4\alpha + 96\rho\sqrt{c}\widetilde{\mathfrak{R}}_{nc}(\mathcal{P}(\mathcal{F}))\log_2(4en^2c^2\rho^2)\cdot\ln\frac{M}{\alpha}\right)$$
$$\leq \frac{12M}{\sqrt{n}} + 96\rho\sqrt{c}\widetilde{\mathfrak{R}}_{nc}(\mathcal{P}(\mathcal{F}))(1 + \log_2(4en^2c^2\rho^2)\cdot\ln\frac{M}{24\rho\sqrt{c}\widetilde{\mathfrak{R}}_{nc}(\mathcal{P}(\mathcal{F}))})$$
$$\text{(Choose } \alpha = 24\rho\sqrt{c}\widetilde{\mathfrak{R}}_{nc}(\mathcal{P}(\mathcal{F})))$$
$$\leq \frac{12M}{\sqrt{n}} + 96\rho\sqrt{c}\widetilde{\mathfrak{R}}_{nc}(\mathcal{P}(\mathcal{F}))(1 + \log_2(4en^2c^2\rho^2)\cdot\ln\frac{M\sqrt{n}}{\rho B}). \quad \text{(Use inequality (7))}$$

### A.5.2. PROOF OF THEOREM 6.3

**Proof Sketch**: First, for the induced surrogate loss for Macro-Averaged AUC, by using the U-process technique, we define the empirical Rademacher complexity of a loss function space associated with the multi-label learning class $\mathcal{F}$ over the set of i.i.d disjoint positive and negative sample pairs for each $j$-th label, then with two-sided multiplicative Chernoff bound, the

generalization error can be bounded by $\hat{\Re}_{D_0}(\mathcal{L}_M) = \mathbb{E}_{\epsilon}\left[\sup_{\boldsymbol{f}\in\mathcal{F}} \frac{1}{r_0}\sum_{i=1}^{r_0}\frac{1}{c}\sum_{j=1}^{c}\epsilon_i\ell_b\left(f_j(\boldsymbol{x}_i^{j+}) - f_j(\boldsymbol{x}_i^{j-})\right)\right]$. Second, combining with Lemma 6.2 and the Lipschitz continuity with respect to the $\ell_\infty$ norm of the induced surrogate loss for Macro-Averaged AUC, we have $\hat{\Re}_{D_0}(\mathcal{L}_M) \leq \frac{12M}{\sqrt{r_0}} + 96\rho\sqrt{c}\widetilde{\Re}_{r_0c}(\mathcal{P}(\mathcal{F}))(1 + \log_2(4er_0^2c^2\rho^2)\cdot\ln\frac{M\sqrt{r_0}}{\rho B})$. Finally, we upper bound the worst-case Rademacher complexity $\widetilde{\Re}_{r_0c}(\mathcal{P}(\mathcal{F})) \leq \frac{2B}{\sqrt{r_0c}}$, the desired bound can be derived.

Rademacher complexity has proved to be a powerful data-dependent measure of hypothesis space complexity. However, since Macro-Averaged AUC involves pairwise functions, a sequence of pairs of i.i.d. individual observation in (2) is no longer independent, which makes standard techniques in the i.i.d case for traditional Rademacher complexity inapplicable. We convert the non-sum-of-i.i.d pairwise function to a sum-of-i.i.d form by using permutations in U-process (Clémençon et al., 2008; Zhang & Zhang, 2024a).

According to the definitions of the empirical risk w.r.t. Macro-Averaged AUC and the induced surrogate loss for Macro-Averaged AUC in (2) and Subsection 3.2, for the induced surrogate loss for Macro-Averaged AUC, we define the empirical Rademacher complexity of a loss function space associated with the multi-label learning class $\mathcal{F}$ over the set of i.i.d disjoint positive and negative sample pairs for the $j$-th label as follows:

$$\hat{\Re}_{D_j}(\mathcal{L}_M) = \mathbb{E}_{\epsilon}\left[\sup_{\boldsymbol{f}\in\mathcal{F}} \frac{1}{r_j}\sum_{i=1}^{r_j}\epsilon_i\ell_M(\boldsymbol{f}(\boldsymbol{x}_i^{j+}, \boldsymbol{x}_i^{j-}))\right] = \mathbb{E}_{\epsilon}\left[\sup_{\boldsymbol{f}\in\mathcal{F}} \frac{1}{r_j}\sum_{i=1}^{r_j}\frac{1}{c}\sum_{j=1}^{c}\epsilon_i\ell_b\left(f_j(\boldsymbol{x}_i^{j+}) - f_j(\boldsymbol{x}_i^{j-})\right)\right], \quad (11)$$

where each $\epsilon_i$ is an independent Rademacher random variable. The corresponding expected Rademacher complexity is defined as $\Re_{r_j}(\mathcal{L}_M) = \mathbb{E}_{D_j}\hat{\Re}_{D_j}(\mathcal{L}_M)$. We denote $\sup_{\boldsymbol{f}\in\mathcal{F}}\left|\frac{1}{r_j}\sum_{i=1}^{r_j}\epsilon_i\ell_M(\boldsymbol{f}(\boldsymbol{x}_i^{j+}, \boldsymbol{x}_i^{j-}))\right|$ as $R_{\mathcal{L}_M}^{D_j}$.

According to the symmetrization technique, we can obtain

$$\mathbb{E}_{D_j}\psi\left(\sup_{\boldsymbol{f}\in\mathcal{F}}\left|R(\boldsymbol{f}) - \widehat{R}_D(\boldsymbol{f})\right|\right)$$

$$\leq \mathbb{E}_{D_j}\psi\left(\sup_{\boldsymbol{f}\in\mathcal{F}}\left|\frac{1}{|X_j^+||X_j^-|}\sum_{\substack{\boldsymbol{x}_i^{j+}\in X_j^+ \\ \boldsymbol{x}_k^{j-}\in X_j^-}}\left(\ell_M(\boldsymbol{f}(\boldsymbol{x}_i^{j+}, \boldsymbol{x}_k^{j-})) - R(\boldsymbol{f})\right)\right|\right)$$

$$\leq \mathbb{E}_{D_j}\psi\left(\sup_{\boldsymbol{f}\in\mathcal{F}}\left|\frac{1}{s_j!\cdot t_j!}\sum_{\pi_{j+},\pi_{j-}}\frac{1}{r_j}\sum_{i=1}^{r_j}\left(\ell_M(\boldsymbol{f}(\boldsymbol{x}_{\pi_{j+}(i)}^{j+}, \boldsymbol{x}_{\pi_{j-}(i)}^{j-})) - R(\boldsymbol{f})\right)\right|\right)$$

$$\leq \mathbb{E}_{D_j}\psi\left(\sup_{\boldsymbol{f}\in\mathcal{F}}\frac{1}{s_j!\cdot t_j!}\sum_{\pi_{j+},\pi_{j-}}\left|\frac{1}{r_j}\sum_{i=1}^{r_j}\left(\ell_M(\boldsymbol{f}(\boldsymbol{x}_{\pi_{j+}(i)}^{j+}, \boldsymbol{x}_{\pi_{j-}(i)}^{j-})) - R(\boldsymbol{f})\right)\right|\right)$$

$$\leq \mathbb{E}_{D_j}\psi\left(\frac{1}{s_j!\cdot t_j!}\sum_{\pi_{j+},\pi_{j-}}\sup_{\boldsymbol{f}\in\mathcal{F}}\left|\frac{1}{r_j}\sum_{i=1}^{r_j}\left(\ell_M(\boldsymbol{f}(\boldsymbol{x}_{\pi_{j+}(i)}^{j+}, \boldsymbol{x}_{\pi_{j-}(i)}^{j-})) - R(\boldsymbol{f})\right)\right|\right) \quad (\psi \text{ is nondecreasing})$$

$$\leq \frac{1}{s_j!\cdot t_j!}\sum_{\pi_{j+},\pi_{j-}}\mathbb{E}_{D_j}\psi\left(\sup_{\boldsymbol{f}\in\mathcal{F}}\left|\frac{1}{r_j}\sum_{i=1}^{r_j}\left(\ell_M(\boldsymbol{f}(\boldsymbol{x}_{\pi(i)}^{j+}, \boldsymbol{x}_{\pi(i)}^{j-})) - R(\boldsymbol{f})\right)\right|\right) \quad (\text{Jensen's inequality})$$

$$\leq \mathbb{E}_{D_j}\psi\left(\sup_{\boldsymbol{f}\in\mathcal{F}}\left|\frac{1}{r_j}\sum_{i=1}^{r_j}\left(\ell_M(\boldsymbol{f}(\boldsymbol{x}_i^{j+}, \boldsymbol{x}_i^{j-})) - R(\boldsymbol{f})\right)\right|\right)$$

$$\leq \mathbb{E}_{D_j}\psi\left(\sup_{\boldsymbol{f}\in\mathcal{F}}\left|\frac{1}{r_j}\sum_{i=1}^{r_j}\left(\ell_M(\boldsymbol{f}(\boldsymbol{x}_i^{j+}, \boldsymbol{x}_i^{j-})) - \mathbb{E}_{D_j}\ell_M(\boldsymbol{f}(\boldsymbol{x}_i^{j+}, \boldsymbol{x}_i^{j-}))\right)\right|\right)$$

$$\leq \mathbb{E}_{D_j} \psi \left( \sup_{\boldsymbol{f} \in \mathcal{F}} \left| \frac{1}{r_j} \sum_{i=1}^{r_j} \left( \ell_M(\boldsymbol{f}(\boldsymbol{x}_i^{j+}, \boldsymbol{x}_i^{j-})) - \mathbb{E}_{D_j'} \ell_M(\boldsymbol{f}(\boldsymbol{x}_i^{j+'}, \boldsymbol{x}_i^{j-'})) \right) \right| \right)$$

($D_j'$ is the set of samples with the same distribution as $D_j$)

$$\leq \mathbb{E}_{D_j, D_j'} \psi \left( \sup_{\boldsymbol{f} \in \mathcal{F}} \left| \frac{1}{r_j} \sum_{i=1}^{r_j} \left( \ell_M(\boldsymbol{f}(\boldsymbol{x}_i^{j+}, \boldsymbol{x}_i^{j-})) - \ell_M(\boldsymbol{f}(\boldsymbol{x}_i^{j+'}, \boldsymbol{x}_i^{j-'})) \right) \right| \right) \quad \text{(Jensen's inequality)}$$

$$\leq \mathbb{E}_{D_j, D_j', \boldsymbol{\epsilon}} \psi \left( \sup_{\boldsymbol{f} \in \mathcal{F}} \left| \frac{1}{r_j} \sum_{i=1}^{r_j} \epsilon_i \left( \ell_M(\boldsymbol{f}(\boldsymbol{x}_i^{j+}, \boldsymbol{x}_i^{j-})) - \ell_M(\boldsymbol{f}(\boldsymbol{x}_i^{j+'}, \boldsymbol{x}_i^{j-'})) \right) \right| \right)$$

$$\leq \frac{1}{2} \mathbb{E}_{D_j, \boldsymbol{\epsilon}} \psi \left( 2 \sup_{\boldsymbol{f} \in \mathcal{F}} \left| \frac{1}{r_j} \sum_{i=1}^{r_j} \epsilon_i \ell_M(\boldsymbol{f}(\boldsymbol{x}_i^{j+}, \boldsymbol{x}_i^{j-})) \right| \right) + \frac{1}{2} \mathbb{E}_{D_j', \boldsymbol{\epsilon}} \psi \left( 2 \sup_{\boldsymbol{f} \in \mathcal{F}} \left| \frac{1}{r_j} \sum_{i=1}^{r_j} -\epsilon_i \ell_M(\boldsymbol{f}(\boldsymbol{x}_i^{j+'}, \boldsymbol{x}_i^{j-'})) \right| \right)$$

$$\leq \mathbb{E}_{D_j, \boldsymbol{\epsilon}} \psi \left( 2 \sup_{\boldsymbol{f} \in \mathcal{F}} \left| \frac{1}{r_j} \sum_{i=1}^{r_j} \epsilon_i \ell_M(\boldsymbol{f}(\boldsymbol{x}_i^{j+}, \boldsymbol{x}_i^{j-})) \right| \right) \quad \text{($D_j'$ with the same distribution as $D_j$)}.$$

Hence, we have

$$\mathbb{E}_{D_j} \psi \left( \sup_{\boldsymbol{f} \in \mathcal{F}} \left| R(\boldsymbol{f}) - \widehat{R}_D(\boldsymbol{f}) \right| \right) \leq \mathbb{E}_{D_j, \boldsymbol{\epsilon}} \psi \left( 2 R_{\mathcal{L}_M}^{D_j} \right). \tag{12}$$

Since the maximum difference caused by replacing one element in $D_j$ or $\epsilon_i$ is $\frac{2M}{r_j}$, according to McDiarmid's inequality, we have

$$P(|R_{\mathcal{L}_M}^{D_j} - \Re_{r_j}(\mathcal{L}_M)| \geq \varepsilon) \leq 2 e^{-\frac{\varepsilon^2 r_j}{2M^2}}.$$

Then, according to the tail bound for sub-Gaussian random variables and Theorem 2.1 in (Boucheron et al., 2013), $R_{\mathcal{L}_M}^{D_j}$ is a sub-Gaussian random variable with variance proxy $16 \frac{M^2}{r_j}$. With the definition of the sub-Gaussian random variable, we have

$$\mathbb{E}_{D_j, \boldsymbol{\epsilon}} e^{t R_{\mathcal{L}_M}^{D_j}} \leq e^{t \Re_{r_j}(\mathcal{L}_M) + \frac{8 t^2 M^2}{r_j}}, \ \forall t > 0. \tag{13}$$

Then, for any $\varepsilon > 0$, we have

$$P \left( \sup_{\boldsymbol{f} \in \mathcal{F}} \left| R(\boldsymbol{f}) - \widehat{R}_D(\boldsymbol{f}) \right| \geq \varepsilon \right)$$

$$= P \left( e^{t \sup_{\boldsymbol{f} \in \mathcal{F}} |R(\boldsymbol{f}) - \widehat{R}_D(\boldsymbol{f})|} \geq e^{t\varepsilon} \right)$$

$$\leq \frac{\mathbb{E}_{D_j} e^{(t \sup_{\boldsymbol{f} \in \mathcal{F}} |R(\boldsymbol{f}) - \widehat{R}_D(\boldsymbol{f})|)}}{e^{t\varepsilon}} \quad \text{(Use Markov's Inequality)}$$

$$\leq \frac{\mathbb{E}_{D_j, \boldsymbol{\epsilon}} e^{2t R_{\mathcal{L}_M}^{D_j}}}{e^{t\varepsilon}} \quad \text{(Use inequality (12) with $\psi(x) = e^{tx}$)}$$

$$\leq \frac{e^{2t \Re_{r_j}(\mathcal{L}_M) + \frac{32 t^2 M^2}{r_j}}}{e^{t\varepsilon}} \quad \text{(Use inequality (13))}.$$

We set $\frac{e^{2t \Re_{r_j}(\mathcal{L}_M) + \frac{32 t^2 M^2}{r_j}}}{e^{t\varepsilon}} = \delta$, then we have $\varepsilon = 2\Re_{r_j}(\mathcal{L}_M) + \frac{32 t M^2}{r_j} + \frac{\ln \frac{1}{\delta}}{t}$.

Hence, we upper bound the term with probability at least $1 - \delta$:

$$\sup_{\boldsymbol{f} \in \mathcal{F}} \left| R(\boldsymbol{f}) - \widehat{R}_D(\boldsymbol{f}) \right| \leq 2\Re_{r_j}(\mathcal{L}_M) + \frac{32 t M^2}{r_j} + \frac{\ln \frac{1}{\delta}}{t}$$

$$\leq 2\Re_{r_j}(\mathcal{L}_M) + 8M \sqrt{\frac{2 \ln \frac{1}{\delta}}{r_j}}. \tag{14}$$

Next, we transform $\Re_{r_j}(\mathcal{L}_M)$ in inequality (14) into $\hat{\Re}_{D_j}(\mathcal{L}_M)$, according to McDiarmid's inequality, it is easy to obtain that the following holds with probability at least $1 - \delta$:

$$\Re_{r_j}(\mathcal{L}_M) \leq \hat{\Re}_{D_j}(\mathcal{L}_M) + M\sqrt{\frac{\ln\frac{1}{\delta}}{2r_j}}. \tag{15}$$

Combining inequalities (14), (15), and Union bound Inequality, we have the following holds with probability at least $1 - \delta$:

$$R(\boldsymbol{f}) - \widehat{R}_D(\boldsymbol{f}) \leq 2\hat{\Re}_{D_j}(\mathcal{L}_M) + 2M\sqrt{\frac{\ln\frac{2}{\delta}}{2r_j}} + 8M\sqrt{\frac{2\ln\frac{2}{\delta}}{r_j}}. \tag{16}$$

Next, we incorporate an additional Chernoff-type argument to obtain a bound that does not involve any random quantities.

First, since $s_j \sim \text{Binomial}(n, p_j)$, we have $\mathbb{E} = np_j$. With the two-sided multiplicative Chernoff bound, we have

$$P\left(|s_j - np_j| \geq rnp_j\right) \leq 2e^{-\frac{r^2 np_j}{3}}, \; \forall r \in (0, 1).$$

Then, we have

$$
\begin{aligned}
&P\left(|\frac{s_j}{n} - p_j| \geq rp_j\right) \\
\leq& P\left(\cup_j\{|\frac{s_j}{n} - p_j| \geq rp_j\}\right) \\
\leq& \sum_j P\left(|\frac{s_j}{n} - p_j| \geq rp_j\right) \quad \text{(Use Union Bound Inequality)} \\
\leq& \sum_j 2e^{-\frac{r^2 np_j}{3}} \\
\leq& 2ce^{-\frac{r^2 n \min_j p_j}{3}}.
\end{aligned}
$$

We set $2ce^{-\frac{r^2 n \min_j p_j}{3}} = \delta$, then we have $r = \sqrt{\frac{3\ln\frac{2c}{\delta}}{n\min_j p_j}}$. Hence, the following holds with probability at least $1 - \delta$:

$$|\frac{s_j}{n} - p_j| \leq p_j\sqrt{\frac{3\ln\frac{2c}{\delta}}{n\min_j p_j}}.$$

Solving the above inequality yields $s_j \geq np_j(1 - \sqrt{\frac{3\ln\frac{2c}{\delta}}{n\min_j p_j}})$. Similarly, we have $|\frac{t_j}{n} - (1 - p_j)| \leq (1 - p_j)\sqrt{\frac{3\ln\frac{2c}{\delta}}{n\min_j p_j}}$, and solving this inequality yields $t_j \geq n(1 - p_j)(1 - \sqrt{\frac{3\ln\frac{2c}{\delta}}{n\min_j p_j}})$.

Hence, we have the following holds with probability at least $1 - \delta$:

$$r_j = \min\{s_j, t_j\} \geq \min\{np_j(1 - \sqrt{\frac{3\ln\frac{2c}{\delta}}{n\min_j p_j}}), n(1 - p_j)(1 - \sqrt{\frac{3\ln\frac{2c}{\delta}}{n\min_j p_j}})\}. \tag{17}$$

In order to ensure that for every $j$-th label, disjoint positive and negative sample pairs can be constructed, we need to derive the lower bound for $\min_j\{r_j\}$ to obtain the number of disjoint positive and negative sample pairs that can be constructed for every $j$-th label. Since

$$
\begin{aligned}
r_j \geq& \min_j\{r_j\} = \min_j\{\min\{s_j, t_j\}\} \\
\geq& \min\{\min_j\{np_j(1 - \sqrt{\frac{3\ln\frac{2c}{\delta}}{n\min_j p_j}})\}, \min_j\{n(1 - p_j)(1 - \sqrt{\frac{3\ln\frac{2c}{\delta}}{n\min_j p_j}})\}\}
\end{aligned}
$$

$$= \min\{n \min_j p_j(1 - \sqrt{\frac{3 \ln \frac{2c}{\delta}}{n \min_j p_j}}), n \min_j(1 - p_j)(1 - \sqrt{\frac{3 \ln \frac{2c}{\delta}}{n \min_j p_j}})\}$$

$$\geq \min\{\frac{1}{2}n \min_j p_j, \frac{1}{2}n \min_j(1 - p_j)\} \quad \text{(Assume that } n \geq \frac{12 \ln \frac{2c}{\delta}}{\min_j p_j})$$

$$= \frac{1}{2}n \min\{\min_j p_j, \min_j(1 - p_j)\}$$

$$:= \frac{1}{2}r_0 \quad \text{(Define } r_0 = n \min\{\min_j p_j, \min_j(1 - p_j)\}),$$

then we have $r_j \geq \frac{1}{2}r_0$ with probability at least $1 - \delta$.

Combining the inequality (16) and Union bound Inequality, we have the following holds with probability at least $1 - \delta$:

$$R(\boldsymbol{f}) - \widehat{R}_D(\boldsymbol{f})$$

$$\leq 2\sqrt{2}\hat{\Re}_{D_0}(\mathcal{L}_M) + 2M\sqrt{\frac{\ln \frac{4}{\delta}}{r_0}} + 16M\sqrt{\frac{\ln \frac{4}{\delta}}{r_0}}$$

$$= 2\sqrt{2}\hat{\Re}_{D_0}(\mathcal{L}_M) + 18M\sqrt{\frac{\ln \frac{4}{\delta}}{r_0}}. \tag{18}$$

Then, according to Lemma 6.2, we have

$$\hat{\Re}_{D_0}(\mathcal{L}_M) \leq \frac{12M}{\sqrt{r_0}} + 96\rho\sqrt{c}\widetilde{\Re}_{r_0c}(\mathcal{P}(\mathcal{F}))(1 + \log_2(4er_0^2c^2\rho^2) \cdot \ln \frac{M\sqrt{r_0}}{\rho B}).$$

We then upper bound the worst-case Rademacher complexity $\widetilde{\Re}_{r_0c}(\mathcal{P}(\mathcal{F}))$ as the following:

$$\widetilde{\Re}_{r_0c}(\mathcal{P}(\mathcal{F}))$$

$$= \sup_{[c] \times D' \in [c] \times \mathcal{X}^{r_0}} \hat{\Re}_{[c] \times D'}(\mathcal{P}(\mathcal{F}))$$

$$= \sup_{[c] \times D' \in [c] \times \mathcal{X}^{r_0}} \mathbb{E}_{\boldsymbol{\epsilon}} \left[ \sup_{p_j(\boldsymbol{f}(\boldsymbol{x}_i)) \in \mathcal{P}(\mathcal{F})} \frac{1}{r_0c} \sum_{i=1}^{r_0} \sum_{j=1}^{c} \epsilon_{ij} p_j(\boldsymbol{f}(\boldsymbol{x}_i, \boldsymbol{x}_i')) \right]$$

$$= \sup_{[c] \times D' \in [c] \times \mathcal{X}^{r_0}} \mathbb{E}_{\boldsymbol{\epsilon}} \left[ \sup_{f_j \in \mathcal{F}_j} \frac{1}{r_0c} \sum_{i=1}^{r_0} \sum_{j=1}^{c} \epsilon_{ij} (f_j(\boldsymbol{x}_i) - f_j(\boldsymbol{x}_i')) \right]$$

$$= \sup_{\|\phi_j(\boldsymbol{x}_i)\| \leq A : i \in [r_0], j \in [c]} \frac{1}{r_0c} \mathbb{E}_{\boldsymbol{\epsilon}} \left[ \sup_{\|\boldsymbol{w}_j\| \leq \Lambda} \sum_{i=1}^{r_0} \sum_{j=1}^{c} \epsilon_{ij} \langle \boldsymbol{w}_j, \phi_j(\boldsymbol{x}_i) - \phi_j(\boldsymbol{x}_i') \rangle \right]$$

$$= \sup_{\|\phi_j(\boldsymbol{x}_i)\| \leq A : i \in [r_0], j \in [c]} \frac{\Lambda}{r_0c} \mathbb{E}_{\boldsymbol{\epsilon}} \| \sum_{i=1}^{r_0} \sum_{j=1}^{c} \epsilon_{ij} (\phi_j(\boldsymbol{x}_i) - \phi_j(\boldsymbol{x}_i')) \|$$

$$\leq \sup_{\|\phi_j(\boldsymbol{x}_i)\| \leq A : i \in [r_0], j \in [c]} \frac{2\Lambda}{r_0c} \mathbb{E}_{\boldsymbol{\epsilon}} \| \sum_{i=1}^{r_0} \sum_{j=1}^{c} \epsilon_{ij} \phi_j(\boldsymbol{x}_i) \|$$

$$\leq \sup_{\|\phi_j(\boldsymbol{x}_i)\| \leq A : i \in [r_0], j \in [c]} \frac{2\Lambda}{r_0c} \left[ \mathbb{E}_{\boldsymbol{\epsilon}} \| \sum_{i=1}^{r_0} \sum_{j=1}^{c} \epsilon_{ij} \phi_j(\boldsymbol{x}_i) \|^2 \right]^{\frac{1}{2}} \quad \text{(Use Jensen's Inequality)}$$

$$\leq \sup_{\|\phi_j(\boldsymbol{x}_i)\| \leq A : i \in [r_0], j \in [c]} \frac{2\Lambda}{r_0c} \left[ \sum_{i=1}^{r_0} \sum_{j=1}^{c} \|\phi_j(\boldsymbol{x}_i)\|^2 \right]^{\frac{1}{2}} \leq \frac{2\Lambda A}{\sqrt{r_0c}} := \frac{2B}{\sqrt{r_0c}}. \quad \text{(Use Lemma A.2)} \tag{19}$$

Then, we have

$$\hat{\Re}_{D_0}(\mathcal{L}_M) \leq \frac{12M}{\sqrt{r_0}} + 96\rho\sqrt{c}\widetilde{\Re}_{r_0 c}(\mathcal{P}(\mathcal{F}))(1 + \log_2(4er_0^2 c^2 \rho^2) \cdot \ln\frac{M\sqrt{r_0}}{\rho B})$$

$$\leq \frac{12M}{\sqrt{r_0}} + \frac{192\rho B}{\sqrt{r_0}}(1 + \log_2(4er_0^2 c^2 \rho^2) \cdot \ln\frac{M\sqrt{r_0}}{\rho B}).$$

Combining with (18), then

$$R(\boldsymbol{f}) - \widehat{R}_D(\boldsymbol{f}) \leq \frac{24\sqrt{2}M}{\sqrt{r_0}} + \frac{384\sqrt{2}\rho B}{\sqrt{r_0}}(1 + \log_2(4er_0^2 c^2 \rho^2) \cdot \ln\frac{M\sqrt{r_0}}{\rho B}) + 18M\sqrt{\frac{\ln\frac{4}{\delta}}{r_0}}.$$

### A.5.3. PROOF OF LEMMA 6.6

We first derive the relationship between the empirical $\ell_2$ norm covering number $\mathcal{N}_2(\epsilon, \mathcal{L}_M, D)$ and the empirical $\ell_\infty$ norm covering number $\mathcal{N}_\infty(\epsilon, \mathcal{P}(\mathcal{F}), [c] \times D)$ by the smoothness of base loss functions.

For the dataset $D = \{(\boldsymbol{x}_1, \boldsymbol{y}_1), \ldots, (\boldsymbol{x}_n, \boldsymbol{y}_n)\}$ with $n$ i.i.d. examples:

$$\sqrt{\frac{1}{n}\sum_{i=1}^{n}\left(\ell_M(\boldsymbol{f}(\boldsymbol{x}_i, \boldsymbol{x}_i'), \boldsymbol{y}) - \ell_M(\boldsymbol{f}'(\boldsymbol{x}_i, \boldsymbol{x}_i'), \boldsymbol{y})\right)^2}$$

$$= \sqrt{\frac{1}{n}\sum_{i=1}^{n}\left(\frac{1}{c}\sum_{j=1}^{c}\ell_b\left(f_j(\boldsymbol{x}_i) - f_j(\boldsymbol{x}_i')\right) - \frac{1}{c}\sum_{j=1}^{c}\ell_b\left(f_j'(\boldsymbol{x}_i) - f_j'(\boldsymbol{x}_i')\right)\right)^2}$$

$$\leq \sqrt{\frac{1}{n}\sum_{i=1}^{n}\frac{1}{c}\sum_{j=1}^{c}\left(\ell_b\left(f_j(\boldsymbol{x}_i) - f_j(\boldsymbol{x}_i')\right) - \ell_b\left(f_j'(\boldsymbol{x}_i) - f_j'(\boldsymbol{x}_i')\right)\right)^2} \quad \text{(Use Jensen's Inequality)}$$

$$\leq \sqrt{\frac{1}{n}\sum_{i=1}^{n}\frac{1}{c}\sum_{j=1}^{c}6\gamma(\ell_b\left(f_j(\boldsymbol{x}_i) - f_j(\boldsymbol{x}_i')\right) + \ell_b\left(f_j'(\boldsymbol{x}_i) - f_j'(\boldsymbol{x}_i')\right))(f_j(\boldsymbol{x}_i) - f_j(\boldsymbol{x}_i') - f_j'(\boldsymbol{x}_i) + f_j'(\boldsymbol{x}_i'))^2}$$

(Use Lemma A.1)

$$\leq \sqrt{\frac{6\gamma}{n}\sum_{i=1}^{n}\frac{1}{c}\sum_{j=1}^{c}(\ell_b\left(f_j(\boldsymbol{x}_i) - f_j(\boldsymbol{x}_i')\right) + \ell_b\left(f_j'(\boldsymbol{x}_i) - f_j'(\boldsymbol{x}_i')\right))}\sqrt{\max_{i\in[n],j\in[c]}(f_j(\boldsymbol{x}_i) - f_j(\boldsymbol{x}_i) - f_j'(\boldsymbol{x}_i) + f_j'(\boldsymbol{x}_i))^2}$$

$$\leq \sqrt{12\gamma M}\max_{i\in[n],j\in[c]}|f_j(\boldsymbol{x}_i) - f_j(\boldsymbol{x}_i') - f_j'(\boldsymbol{x}_i) + f_j'(\boldsymbol{x}_i')| \quad \text{(Use Assumption 4.1)}$$

$$\leq \sqrt{12\gamma M}\max_{i\in[n],j\in[c]}|(f_j(\boldsymbol{x}_i, \boldsymbol{x}_i') - f_j'(\boldsymbol{x}_i, \boldsymbol{x}_i')|$$

$$\leq \sqrt{12\gamma M}\max_{i}\max_{j}|p_j(\boldsymbol{f}(\boldsymbol{x}_i, \boldsymbol{x}_i')) - p_j(\boldsymbol{f}'(\boldsymbol{x}_i, \boldsymbol{x}_i'))|. \quad \text{(The definition of the projection function class } \mathcal{P}(\mathcal{F}))$$

Then, according to the definitions of the empirical $\ell_2$ and $\ell_\infty$ covering number, we have that an empirical $\ell_\infty$ cover of $\mathcal{P}(\mathcal{F})$ at radius $\epsilon/\sqrt{12\gamma M}$ is also an empirical $\ell_2$ cover of the loss function space $\mathcal{L}_M$ at radius $\epsilon$, and we can conclude that:

$$\mathcal{N}_2\left(\epsilon, \mathcal{L}_M, D\right) \leq \mathcal{N}_\infty\left(\frac{\epsilon}{\sqrt{12\gamma M}}, \mathcal{P}(\mathcal{F}), [c] \times D\right).$$

The following proof process is similar to the proof of Lemma 4.3. Hence, we have:

$$\hat{\Re}_D(\mathcal{L}_M) \leq \frac{12M}{\sqrt{n}} + 192\sqrt{3c\gamma M}\widetilde{\Re}_{nc}(\mathcal{P}(\mathcal{F}))(1 + \log_2(48en^2 c^2 \gamma M) \cdot \ln\frac{\sqrt{nM}}{\sqrt{\gamma}B}).$$

A.5.4. PROOF OF THEOREM 6.7

The overall proof process is similar to the proof of Theorem 6.3, the difference is that according to Lemma 6.6 we have

$$\hat{\Re}_{D_0}(\mathcal{L}_M) \leq \frac{12M}{\sqrt{r_0}} + 192\sqrt{3c\gamma M}\widetilde{\Re}_{r_0c}(\mathcal{P}(\mathcal{F}))(1 + \log_2(48er_0^2c^2\gamma M) \cdot \ln\frac{\sqrt{r_0M}}{\sqrt{\gamma}B}).$$

Combined with the upper bound of the worst-case Rademacher complexity $\widetilde{\Re}_{r_0c}(\mathcal{P}(\mathcal{F}))$ in inequality (19), we have

$$\hat{\Re}_{D_0}(\mathcal{L}_M) \leq \frac{12M}{\sqrt{r_0}} + \frac{384\sqrt{3\gamma M}B}{\sqrt{r_0}}(1 + \log_2(48er_0^2c^2\gamma M) \cdot \ln\frac{\sqrt{r_0M}}{\sqrt{\gamma}B}).$$

Combining with (18), then

$$R(\boldsymbol{f}) - \widehat{R}_D(\boldsymbol{f}) \leq \frac{24\sqrt{2}M}{\sqrt{r_0}} + \frac{768\sqrt{6\gamma M}B}{\sqrt{r_0}}(1 + \log_2(48er_0^2c^2\gamma M) \cdot \ln\frac{\sqrt{r_0M}}{\sqrt{\gamma}B}) + 18M\sqrt{\frac{\ln\frac{4}{\delta}}{r_0}}.$$

