# OpenReview forum: "Tight and Fast Bounds for Multi-Label Learning"
_ICML.cc/2025/Conference — ICML 2025 poster_

### Official Review · Reviewer_Wn57 · 2025-03-05

**Overall Recommendation:** 3

**Summary:**

This paper provide general theoretical guarantees for the generalization of multi-label learning. By developing and leveraging a novel vector-contraction inequalities for smooth base losses, the author induces tight generalization bounds for multi-label learning that have no dependency on the number of labels. Besides, the author develops novel local vector-contraction inequalities for smooth base losses, it can have a bound with faster convergence rate. A tight generalization bounds with no dependency on the number of labels is derived for Macro-Averaged AUC by considering both Lipschitz continuity and smoothness of base loss functions.

**Claims And Evidence:**

Yes

**Essential References Not Discussed:**

N/A

**Experimental Designs Or Analyses:**

N/A

**Methods And Evaluation Criteria:**

Yes

**Other Comments Or Suggestions:**

N/A

**Other Strengths And Weaknesses:**

Strengths

- This paper extend the previous work and introduce novel vector-contraction inequalities for smooth base losses.
- This paper achieves the SoTA generalization bounds and also reveal the relationship between Macro-Averaged AUC and class-imbalance.
- Compared with prior methods, this apper removes the dependency on label count and achieves faster bounds.

Weaknesses

- This paper is highly technical and not reader-friendly, it would be great to show some intuitive explanations or some demos to help the reader understand the key ideas.

**Questions For Authors:**

- Any limitations when the results are applied to the real-world multi-label learning scenarios?

- Since there are many related papers mentioned and compared with the derived results, a more clear and explicit comparison like a summary table with the results and limitations can make the contribution clearer.

**Relation To Broader Scientific Literature:**

This paper introduces vector-contraction inequalities for smooth base loss functions, rather than Lipschitz ones. It shows tight generalization bounds for multi-label learning that have no dependency on the number of labels.

**Theoretical Claims:**

Incorrect statements are not observed in the key claims (e.g., tighter bounds for smooth base losses, faster bounds,  tighter bounds for macro-averaged AUC).

---

> ### Author Rebuttal · Authors · 2025-03-31
>
> Thank you for your constructive comments and active interest in helping us improve the quality of the paper.
>
> The following are our responses to the Questions:
>
> **1. Response to Weakness.**
>
> We will add concrete examples of multi-label methods after the definition of the function class to improve readability and practical interpretation. For example, the DNN-based multi-label method named CLIF [1], which proposes to learn label semantics and representations with specific discriminative properties for each class label in a collaborative way, can be expressed in our function class as:
>
> $$
> \phi_j(\boldsymbol{x})= \sigma_{ReLU} \\{ W_5 \cdot \left[  \sigma_{ReLU} (W_4 \boldsymbol{x}) \odot  \sigma_{sig} (W_3 \psi(Y)_j) \right] \\} .
> $$
>
> The label embeddings $\psi(Y)$ can be denoted by $\sigma_{ReLU}(\tilde{A} \sigma_{ReLU} (\tilde{A} Y W_1) W_2)$, where $\tilde{A}$ denote the normalized adjacency matrix with self-connections, $Y$ is the node feature matrix of the label graph, $\sigma_{ReLU}$ is the ReLU activation, $\sigma_{sig}$ is the sigmoid activation, $\odot$ is the Hadamard product, $W_i$ are the parameter metrices, $i \in [5]$.
>
> In addition, a class of multi-label methods based on the strategy of label-specific representation, which facilitates the discrimination of each class label by tailoring its own representations, can be formalized in our function class. For example, the wrapped label-specific representation method [2], which presents a kernelized Lasso-based framework with the constraint of pairwise label correlations for each class label, can be expressed in our function class, where $f_j$ is the kernelized linear model and the constraint $\alpha(\boldsymbol{w}) $ is $\|\boldsymbol{w}_j\|_1 \leq \Lambda$ for any $j \in [c]$, and each label also has the property of sharing which is reflected by the additionally introduced constraint
>
> $\sum_i^c (1- s_{ji}) {\boldsymbol{w}_j}^\top \boldsymbol{w}_i \leq \tau$,
>
> where $s_{ji}$ is the cosine similarity between labels $y_j$ and $y_i$.
>
> Besides, the function class here is applicable to the typical Binary Relevance methods for multi-label learning, where different methods correspond to different nonlinear mappings $\phi_j$.
>
> [1] Collaborative Learning of Label Semantics and Deep Label-Specific Features for Multi-Label Classification, TPAMI 2022.
>
> [2] Multi-Label Classification with Label-Specific Feature Generation: A Wrapped Approach, TPAMI 2022.
>
> In addition, we will also provide additional explanations on the key ideas in the theoretical proof to help readers better understand the ideas. Please refer to our **Response 3 to Reviewer XSTz**.
>
>
> **2. Response to Q1.**
>
> Our theoretical results require that the base loss is bounded, and therefore cannot cover the case where the base loss is cross-entropy loss. In the future, we will further study the bounds with a faster convergence rate for unbounded and Lipschitz base losses. In addition, for some methods involving specific label correlations, when using our theoretical results to analyze these specific methods, it is necessary to introduce additional assumptions induced by these specific label correlations in the generalization analysis, so as to better reveal the impact of these setting-related factors on the bound. However, how to explicitly introduce label correlations in generalization analysis is still a crucial open problem. We will further explore related work in the future. We will incorporate these discussions into the paper in an appropriate manner.
>
> **3. Response to Q2.**
>
> As the reviewer suggested, a better presentation of related work and comparisons between them through a summary table can improve the readability of the paper and make the contribution of our results clearer. We will add a summary table in the revised version from the perspectives of loss function, additional assumption, order of bounds, and reference.

---

> > ### Comment · Reviewer_Wn57 · 2025-04-03
> >
> > Thanks for the clarification! Most of my concerns are well addressed.

---

> > > ### Author Response · Authors · 2025-04-03
> > >
> > > Thank you again for your support.

---

### Official Review · Reviewer_tz9D · 2025-03-06

**Overall Recommendation:** 4

**Summary:**

The paper focuses on the theoretical analysis of multi-label learning, particularly in the context of smooth base loss functions. The authors introduce novel vector-contraction inequalities and derive tighter generalization bounds for multi-label learning with smooth base loss functions. These bounds exhibit improved dependency on the number of labels, reducing it to logarithmic terms, and demonstrate faster convergence rates with respect to the number of examples. The paper also discusses the application of these bounds to various multi-label learning methods, highlighting how these results provide general theoretical guarantees for the generalization performance of multi-label learning, especially for methods with smooth base loss functions.

**Claims And Evidence:**

Yes. Specifically, the authors derive theoretical support for tight generalisation bounds and more efficient convergence rates for smooth basis loss functions in multi-label learning by means of mathematical proofs and local Rademacher complexity analyses. The authors also give a proof of broad applicability.

**Essential References Not Discussed:**

No.I keep up with the literature in this area.

**Experimental Designs Or Analyses:**

No, the paper is purely a theoretical derivation without any numerical experimental analysis.

**Methods And Evaluation Criteria:**

Yes. The theoretical analysis presented in the article, especially the tight generalisation bounds based on the common smoothed basis loss function, directly addresses the generalisation problem in multi-label learning.In addition, the generalisation bounds proposed in the article not only consider the effect of the number of labels on the model, but also improve the traditional Rademacher complexity analysis by localising the Rademacher complexity, providing faster convergence for large-scale datasets.

**Other Comments Or Suggestions:**

1. The subscript on the left side of the inequality in line 313 is incorrect？
2. The inequality in line 1026 to line 1030 are the same. Please check again.

**Other Strengths And Weaknesses:**

1. A new vector contraction inequality is introduced in the article to derive tight generalisation bounds for multi-label learning. the proof method is very rigorous, but in practice, if the label distribution of the datasets does not exactly match the assumptions, it may affect the applicability of the method.
2. Although the introduction of local Rademacher complexity provides an improvement in convergence speed, the estimation of local complexity may face high computational costs, especially when the number of labels and sample size are very large.The correctness and practical feasibility of the proof relies on being able to efficiently estimate the local complexity and this may be challenging on large-scale datasets.

**Questions For Authors:**

1. The universality of generalisation bounds, although theoretically proven, might there be some limitations for some special types of multi-label problems? Such as high-dimensional sparse data or highly unbalanced label scenarios.
2. The article provides a comparison with existing methods and demonstrates clear improvements.However, inter-label correlation may affect the theory in specific cases. For example, is the traditional Lipschitz approach likely to be more advantageous in the case of strong label correlation? How can more experiments be designed to verify the validity of these theories in practical applications.

**Relation To Broader Scientific Literature:**

Earlier generalisation analyses on multi-label learning, e.g. Wu and Zhu (2020) based on the loss function of Lipschitz continuity can derive generalisation bounds, but the part of the bounds that depends on the number of labels is linear.This thesis successfully reduces the effect of the number of labels on the generalisation bounds, in particular so that the generalisation bounds no longer depend on the number of labels but are only related to the logarithmic term, demonstrating a significant theoretical advance. Bartlett et al. (2005) proposed local Rademacher complexity and showed that it can provide faster convergence than traditional Rademacher complexity.In this thesis, the speed of convergence and generalisation bounds in multi-label learning are improved by introducing the local Rademacher complexity, which provides a faster convergence speed than the traditional method, and the effectiveness of this method is verified theoretically. Wu et al. (2023) proposed a generalisation analysis on macroscopic AUC and explored the effect of label imbalance on the results, without delving into how to reduce the dependence of the number of labels on the generalisation bounds.In this thesis, by combining the Lipschitz and smooth basis loss functions, tight generalisation bounds for macroscopic AUC are given and there is no dependence on the number of labels, which provides a new theoretical guarantee to deal with the label imbalance problem.

**Theoretical Claims:**

Yes. 1. A new vector contraction inequality. This proof is based on the existing Rademacher complexity theory and an extension of local Rademacher complexity that rationally simplifies the otherwise complex multi-label learning problem by smoothing the basis loss function. 2. Application of local Rademacher complexity. This proof describes the complexity of samples in multi-label learning more finely by introducing local complexity.

---

> ### Author Rebuttal · Authors · 2025-03-31
>
> Thank you for your constructive comments and active interest in helping us improve the quality of the paper.
>
> The following are our responses to the Questions:
>
> **1. Response to Weakness 1.**
>
> As the reviewer pointed out, when there is some type of label correlation between the labels of the dataset, the label distribution may satisfy some potential constraints, which may correspond to some additional assumptions such as sparsity assumptions or norm regularization constraints. Therefore, when dealing with these specific problems, we need to introduce some additional assumptions to adjust our analysis and explicitly introduce these potential label correlations into the generalization analysis. This is still an open problem and we will further explore related work in the future.
>
> **2. Response to Weakness 2.**
>
> When dealing with large-scale datasets, in practice, one often consider introducing specific strategies in the label space to deal with extremely large number of labels. In generalization analysis, it is shown that introducing effective and general specific strategies is not only an important open problem in theory, but also extremely challenging in practice. Therefore, it is indeed necessary to further introduce effective assumptions to better explicitly analyze the role of key factors in generalization that can effectively deal with challenging problems with large number of labels and large-scale datasets.
>
> **3. Response to Suggestions.**
>
> In line 313, the subscript $S$ in the inequality should be changed to $R$.
>
> In line 1026 to 1030, this is a repeated typo and we will remove one of them.
>
> **4. Response to Q1.**
>
> As we replied above, effective analysis for more specific problems requires further explicit introduction of valid assumptions to reveal the impact of these setting-dependent factors on the bound. For example, for high-dimensional sparse data, one may need to introduce sparsity assumptions into the analysis, thereby inducing bounds that are weakly dependent on the sparsity rate and the number of key labels.
>
> **5. Response to Q2.**
>
> Different types of label correlations have an important impact on generalization analysis. How to explicitly introduce them in generalization analysis is a crucial open problem. Traditional Lipschitz methods do have more advantages since they are easier to satisfy the conditions. For example, for deep models, the smoothness of base losses also involves the boundedness of the second-order derivative of the function, which is often difficult to guarantee in deep models, while Lipschitz continuity is often established. Experimental verification can be considered from two aspects. On the one hand, verify whether the functions selected by the algorithm have a small error and whether the generalization performance of the small error functions is better. On the other hand, verify whether the smoothness of the model function can be guaranteed by some regularization, and explore which regularization-induced inductive biases are more effective for generalization in practice, and promote further theoretical research.
>
> We will incorporate these discussions into the paper in an appropriate manner.

---

### Official Review · Reviewer_ZadH · 2025-03-12

**Overall Recommendation:** 3

**Summary:**

This paper investigates the generalization bound of multi-label loss functions. Specifically, for smooth base loss functions, the authors improve the generalization bounds by removing the dependency on the number of labels $c$. By exploiting local Rademacher complexity, the authors further improve the bound from $\tilde{O}(1/\sqrt{n})$ to $\tilde{O}(1/n)$. In addition, they. also derive tight bounds for Macro-Averaged AUC.

**Claims And Evidence:**

The theoretical claims are.supported by proofs and comparisons with previous works.

**Essential References Not Discussed:**

NA.

**Experimental Designs Or Analyses:**

NA.

**Methods And Evaluation Criteria:**

NA.

**Other Comments Or Suggestions:**

NA.

**Other Strengths And Weaknesses:**

**Strengths.**
1. This paper is overall well written, with sufficient discussions on the differences from previous works.
2. The derived bounds are tighter removing the reliance on the number of classes, while introducing no additional strong assumptions.

**Weakness.**
1. The theoretical results do not cover unbounded base loss functions, e.g. cross entropy loss.
2. Lack of experimental validation of the theoretical results. e.g. does the generalization gap really not depend on $c$? (Perhaps it is not possible to calculate the population risk?)

**Questions For Authors:**

1. Definition 3.3. The fat-shattering dimension seems to depend on the witnesses $s_1,\ldots,s_p$. What are the meanings of these witnesses?
2. Theorem 5.5. The population risk $R(f)$ is bounded by $2\hat{R}_D(f)$ instead of $\hat{R}_D(f)$. Does it mean that this bound is somewhat loose? As $n\to\infty$, the bound becomes $R(f)\leq 2R(f)$, which is not tight.

**Relation To Broader Scientific Literature:**

NA.

**Theoretical Claims:**

I only read the proof sketches, which seem to make sense.

---

> ### Author Rebuttal · Authors · 2025-03-31
>
> Thank you for your constructive comments and active interest in helping us improve the quality of the paper.
>
> The following are our responses to the Questions:
>
> **1. Response to Weakness 1.**
>
> As the reviewer commented, the theoretical results for unbounded base losses need to be further explored in the future to develop new theoretical techniques to cover this situation. Under the new theoretical analysis method, although the smoothness of the cross-entropy loss may be difficult to hold for models with large capacity, the Lipschitz continuity of the cross-entropy is often established, i.e., tight bounds with no dependency on $c$ for cross-entropy loss can be obtained. However, improving the convergence rate of its bound with respect to $n$ is still an urgent open problem to be solved. In the future, we will further study the bounds with a faster convergence rate for unbounded and Lipschitz base losses.
>
> **2. Response to Weakness 2.**
>
> Since the $\widetilde{O}(\frac{1}{\sqrt{n}} )$ and $\widetilde{O}(\frac{1}{n} )$ bounds in our theoretical results is independent of the number of labels, **up to logarithmic terms**, the dependency of the bounds here on $c$ is logarithmic. We use $\widetilde{O}$ to omit the logarithmic terms since the logarithmic dependency is very weak. However, there is no contradiction between the theoretical results and empirical intuition. The increase in $c$ will affect the difficulty of learning, but the empirical success of multi-label methods suggests that the increase in $c$ have limited impact on the difficulty of learning, which means that the ideal bound should not be strongly dependent on $c$.
>
> **3. Response to Question 1.**
>
> If there are real numbers $s_1, \ldots, s_p$ such that for each $\delta_1, \ldots, \delta_p \in  \\{-1, +1\\}$ there exists $f \in \mathcal{F}$ with
>  $$
> \delta_i\left(f(\boldsymbol{x}_{i})-s_i\right) \geq \epsilon, \quad \forall i = 1, \ldots, p.
> $$
>  We say that $s_1, \ldots, s_p$ witness the shattering.
>
>
> **4. Response to Question 2.**
>
> The multiplicative factor of 2 comes from the use of Lemma A.8, which can also be understood through its proof process. The result can often be shown through some derivation as follows:
>
> $$
> R(f) \leq  \widehat{R}_{D}(f) +  \widetilde{O}(1/n + \sqrt{R^* / n} ) ,
> $$
>
> where $R^* = \inf_{f\in \mathcal{F}} R(f)$. This means that in the separable case ($R^* =0$), the bound can be improved to a $\widetilde{O}(1/n)$ rate. The multiplicative factors often appear in bounds based on local Rademacher complexity, as also shown in literature [1-3], etc.
>
>
> [1] Local Rademacher Complexity-based Learning Guarantees for Multi-Task Learning, JMLR 2018.
>
> [2] Towards Sharper Generalization Bounds for Structured Prediction, NeurIPS 2021.
>
> [3] Generalization Analysis for Ranking Using Integral Operator, AAAI 2017.

---

> > ### Comment · Reviewer_ZadH · 2025-04-05
> >
> > Thanks for the reply. I have no further questions and decide to keep my rating.

---

> > > ### Author Response · Authors · 2025-04-05
> > >
> > > Thank you again for your support.

---

### Official Review · Reviewer_dgGN · 2025-03-14

**Overall Recommendation:** 2

**Summary:**

By incorporating smoothness assumption, author provides generalization guarantee achieving a tighter bound - independent of c, the number of labels up to log factors, a faster bound - 1/n, and a similar tighter bound for Macro-averaged AUC.

**Claims And Evidence:**

Mostly seems sound, but I have a question.
Zhang & Zhang 2024, states in the introduction, remark 3.8, and remark 3.18 that their bound is also independent of c.
But in this paper, you only mention sqrt(c) factor bound in the related work section for Zhang & Zhang.
Am I missing something? It would be good to state the improvement.

**Essential References Not Discussed:**

I recommend considering discussing the paper,
Busa-Fekete, Róbert, et al. "Regret bounds for multilabel classification in sparse label regimes." Advances in Neural Information Processing Systems 35 (2022), since the paper discusses obtaining fast rates (or ultra fast rates)  in multi-label setting, which is an important contribution of the paper.

**Experimental Designs Or Analyses:**

No experiments.

**Methods And Evaluation Criteria:**

No experiments.

**Other Comments Or Suggestions:**

I think some part of definition 3.2 is cut-off around line 166.

**Other Strengths And Weaknesses:**

I find the paper well grounded.

**Questions For Authors:**

I mentioned by question in "Claims And Evidence" above.

**Relation To Broader Scientific Literature:**

I think this is well discussed in the introduction and related works.

**Theoretical Claims:**

I did not go over the proofs.

---

> ### Author Rebuttal · Authors · 2025-03-31
>
> Thank you for your constructive comments and active interest in helping us improve the quality of the paper.
>
> The following are our responses to the Questions:
>
> **1. Response to the Question in Claims And Evidence.**
>
> Here we mention that the bounds with a square-root dependency on $c$ in literature [1] mainly refers to the results for $\ell_2$ Lipschitz loss in literature [1], i.e., Lemma 3.6 and Theorem 3.7. Our improvement is mainly relative to Theorem 3.7. In the proof of Lemma 3.6, the $n$ in equation (9) should be changed to $nc$. This is a typo. In fact, the results in Lemma 3.6 and Theorem 3.7 require the introduction of an additional $\sqrt{c}$ factor, because they ignore the $\sqrt{c}$ factor in the radius of the empirical $\ell_2$ cover of $\mathcal{P}(\mathcal{F})$. Therefore, a $\sqrt{c}$ factor is missing in Lemma 3.6 and Theorem 3.7, and literature [1] improved the dependency of the bounds on $c$ from linear to square-root in the decoupling case for $\ell_2$ norm Lipschitz losses. Although for $\ell_2$ Lipschitz loss, the bounds in literature [1] is only improved by a factor of $\sqrt{c}$, they are still the tightest results in multi-label learning with $\ell_2$ Lipschitz loss. In addition, for Hamming loss, its Lipschitz constant can induce the tight bounds with no dependency on $c$. We found that the square-root dependency of the bound in [1] on $c$ is inevitable for $\ell_2$ Lipschitz loss, which essentially comes from the $\sqrt{c}$ factor in the radius of the empirical $\ell_2$ cover of the projection function class. We also found that the smoothness of the base loss function can eliminate the $\sqrt{c}$ factor in the radius of the empirical $\ell_2$ cover of the projection function class, so that the tight bounds with no dependency on $c$, up to logarithmic terms, can be derived.
>
> [1] Yi-Fan Zhang, Min-Ling Zhang. "Generalization Analysis for Multi-Label Learning", ICML 2024.
>
>
> **2. Response to the Essential Reference.**
>
> The literature [2] derived tight bounds with a logarithmic dependency on $c$ for Hamming loss with KNN under the smoothness assumption of the regression function and multi-label margin and sparsity assumptions and also derived tight bounds with a logarithmic dependency on $c$ for Precision@$\kappa$ under the margin condition and the smoothness assumption. The margin condition ensures that the obtained bounds with a faster convergence rate. In our work, the local loss function space is the key to obtaining bounds with a faster convergence rate. The smoothness condition with respect to the $\ell_\infty$ norm in literature [2] is a variant of Holder-continuity. We also find that the $\ell_\infty$ norm has a positive effect on obtaining tight bound with a weaker dependency on $c$, i.e., tight bounds with a logarithmic dependency on $c$ can be derived for $\ell_\infty$ Lipschitz losses. However, how to improve the convergence rate of the bounds for Lipschitz losses is still an open problem, which we will further explore in future work. We will incorporate these discussions into the paper in an appropriate manner.
>
> [2] Regret Bounds for Multilabel Classification in Sparse Label Regimes. NeurIPS 2022.
>
> **3. Response to the Suggestion.**
>
> In order to avoid the possibility of truncation in understanding, we will adjust Definition 3.2 to $\ell_p$ norm covering number instead of listing the $\ell_2$ norm and $\ell_\infty$ norm covering number separately in the definition.

---

> > ### Comment · Reviewer_dgGN · 2025-04-03
> >
> > Thank you for the rebuttal. After reading the rebuttal, I feel that comparison to [1] needs to be discussed in detail and verified carefully to convey the main important point of the paper, which should have been stated in the paper. I lower my score.

---

> > > ### Author Response · Authors · 2025-04-03
> > >
> > > Thanks for your efforts in making our work clearer for readers. We compare with [1] in more detail and convey our key points as follows, and we hope our further response will address your concerns.
> > >
> > > Regarding the reviewer's comments "[1] states in the introduction, remark 3.8, and remark 3.18 that their bound is also independent of $c$", Remark 3.8 refers to the bound for $\ell_2$ Lipschitz loss, and Remark 3.18 refers to the bound for $\ell_\infty$ Lipschitz loss. There is no problem that the bound for $\ell_\infty$ Lipschitz loss is independent of $c$. In fact, we have also pointed out in related work that "[1] derived a $\widetilde{O}(1/\sqrt{n})$ bound for $\ell_\infty$ Lipschitz loss...". When discussing bounds for Macro-AUC, we show in Remark 6.4 that our bound is tighter than bound for Macro-AUC in [1], since bound in [1] uses a looser vector-contraction inequality, while we develop a tight vector-contraction inequality for the case where base loss is smooth, which can improve the bound by a factor of $\sqrt{c}$. These discussions are clearly explained in our paper.
> > >
> > > Below we explain more clearly the part that may cause confusion, i.e., the relevant results in Remark 3.8 of [1], which is "the bound with a square-root dependency on $c$ in [1]" described in the related work **for $\ell_2$ Lipschitz loss** in our paper. Please note that when we mention the bound with a square-root dependency on $c$, we emphasize the expression "**for $\ell_2$ Lipschitz loss**". In Remark 3.8 of [1], it is shown that bound for $\ell_2$ Lipschitz loss is independent of $c$. The conflict between these two descriptions is that Step 3 of the proof of Lemma 3.6 in [1] ignores the $\sqrt{c}$ factor in the radius of empirical $\ell_2$ cover of $\mathcal{P}(\mathcal{F})$. Hence, the third inequality below Eqn (10) in Step 3 of [1] should be modified as:
> > >
> > > $$
> > > \inf_{\alpha>0} \left( 4 \alpha+48 \sqrt{c} \mu \sqrt{c} \widetilde{\Re}_{nc}(\mathcal{P}(\mathcal{F})) \log^{\frac{1}{2}} (nc) \cdot T\right),
> > > $$
> > >
> > > $$
> > > \text{(where $T=\int_{\alpha}^M \epsilon^{-1} d\epsilon$)}
> > > $$
> > >
> > > which will introduce an additional $\sqrt{c}$ factor and cause bounds in Lemma 3.6 and Theorem 3.7 to be square-root dependent on $c$. Hence, [1] improved the dependency of bounds on $c$ from linear to square-root in the decoupling case **for $\ell_2$ Lipschitz loss**.
> > >
> > > In fact, we have previously confirmed and reached agreement with the authors of [1] on this issue. This issue does not affect the conclusion of [1] in general, since for Hamming loss, the inverse of the $\sqrt{c}$ factor in its Lipschitz constant can induce tight bounds with no dependency on $c$.
> > >
> > > Our improvement mainly stems from the observation that for $\ell_2$ Lipschitz loss, the square-root dependency of bound in [1] on $c$ is inevitable, which essentially comes from the $\sqrt{c}$ factor in the radius of the empirical $\ell_2$ cover of the projection function class $\mathcal{P}(\mathcal{F})$, i.e., $\frac{\epsilon}{\mu \sqrt{c}}$. After careful analysis, we found that smoothness of base loss can eliminate this $\sqrt{c}$ factor, i.e., $\frac{\epsilon}{ \sqrt{12 \gamma M} }$. In addition, the method based on Sudakov's minoration used in [1] to upper bound the $\ell_2$ norm covering number of the projection function class is no longer applicable here. In our paper, according to the smoothness of base losses, we first derive the relationship between empirical $\ell_{2}$ norm covering number of the loss space and empirical $\ell_\infty$ norm covering number of the projection function class. Then, we show that empirical $\ell_\infty$ norm covering number $\mathcal{N}_{\infty}(\epsilon, \mathcal{P}(\mathcal{F}), [c] \times D)$ can be bounded by worst-case Rademacher complexity of the projection function class by using the fat-shattering dimension as a bridge.
> > >
> > > The above key points and proof ideas can induce a bound independent of $c$. Hence, for the bound with a square-root dependency on $c$ for $\ell_2$ Lipschitz loss in [1], we consider the smoothness of base loss and improve the bound by a factor of $\sqrt{c}$. In addition, the smoothness of base loss combined with the local loss function space allows the development of novel local vector-contraction inequalities, which can induce bounds that not only have a faster convergence rate but also have a weaker dependency on $c$.
> > >
> > > We will incorporate the above discussion into the paper in a suitable way to better convey the main important point of the paper, and we will not ignore the contribution of [1] to the multi-label community, especially the bounds with no dependency on $c$ for $\ell_\infty$ Lipschitz loss, which we objectively point out in the paper. We will objectively describe the relevant results without any negative impact.
> > >
> > > We hope that our response will help further improve your opinion of our contributions. We are eager to hear back from you if you have any feedback or further questions, and we would love to know your updated reviews.

---

### Official Review · Reviewer_XSTz · 2025-03-18

**Overall Recommendation:** 3

**Summary:**

This paper focuses on the problem of multi-label classification, where each instance can be associated with multiple labels simultaneously. The authors derive several generalization bounds for this setting, assuming smooth loss functions. Their analysis relies on standard techniques for characterizing the complexity of function classes, such as local Rademacher complexity.


The core novelty of the paper lies in the development of specific vector contraction inequalities tailored to a particular class of multi-label classifiers. These inequalities are then used to establish generalization bounds. This approach leads to the derivation of slower convergence rates with respect to the sample size. Notably, the authors obtain rates of $\sqrt{c}$ and $c^{3/2}$ for several well-known multi-label losses, including Hamming loss, subset loss, and macro-averaged AUC (Area Under the Curve).

**Claims And Evidence:**

In summary, while the paper presents valuable generalization bounds for multi-label classification, there are several points that need clarification and improvement. Providing concrete examples of the model class, addressing the applicability to cross-entropy loss, clearly explaining the novelty of the vector contraction inequalities, and correcting the error in Theorem 5.5 would significantly strengthen the paper and enhance its accessibility and impact.

**Essential References Not Discussed:**

see comments above

**Experimental Designs Or Analyses:**

No experiments are provided.

**Methods And Evaluation Criteria:**

No empirical evidence is given.

**Other Comments Or Suggestions:**

None.

**Other Strengths And Weaknesses:**

The notation is quite hard to follow. And there are many inconsistency in the notation. Like $\ell_b$ has one argument, and some cases two arguments. And R_nc is not defined, or I have not found it.

**Questions For Authors:**

See above.

**Relation To Broader Scientific Literature:**

Not included a paper which gives actually much better dependence on the number of labels under a more mild  assumption on the function class than that of considered in the submission. Please see: Busa-Fekete et al.: Regret Bounds for Multilabel Classification in Sparse Label Regimes. NeurIPS 2022

**Theoretical Claims:**

*Specific Points and Concerns:*

1) *Model Class Examples:*
* It would be beneficial if the authors could provide concrete examples of state-of-the-art (SOTA) multi-label classification methods that fall within the specific class of classifiers considered in this paper. This would help readers understand the scope and applicability of the theoretical results. Clarifying which existing models align with their defined model class is crucial for practical interpretation.

2) *Smoothness and Cross-Entropy Loss:*
* In practice, a common approach for multi-label classification involves using cross-entropy loss for each label independently as a surrogate loss function. It is essential to clarify whether the authors' definition of "smoothness" encompasses this widely used cross-entropy loss. If it does, this should be explicitly stated. If not, the limitations of the analysis concerning this practical loss function should be discussed.

3) *Novelty of Vector Contraction Inequalities:*
* The paper's primary contribution seems to be the new vector contraction inequalities. However, the explanation of their novelty is lacking. It would greatly improve the paper if the authors highlighted the key insights or "core idea" behind their improved inequalities. What specific techniques or arguments allowed them to derive better bounds compared to existing vector contraction inequalities? Clearly articulating this contribution is essential for the paper's impact.

4) *Error in Theorem 5.5:*
* Theorem 5.5 appears to contain an error. The empirical error term seems to have a multiplicative factor of 2, which is likely incorrect. This should be carefully checked and corrected. Such errors can significantly undermine the credibility of the theoretical results.

---

> ### Author Rebuttal · Authors · 2025-03-31
>
> Thank you for your constructive comments and active interest in helping us improve the quality of the paper.
>
> **1. Response to C1**
>
> We add concrete examples of multi-label learning (MLL) methods after the function class definition to improve readability and practical interpretation, please refer to **Response 1 to Reviewer Wn57** due to character limitation.
>
> **2. Response to C2**
>
> For the case where base loss is cross-entropy loss, our theoretical results do not cover cross-entropy, mainly because cross-entropy loss is not bounded. Next we will further discuss the smoothness of cross-entropy. When the model is a linear classifier, smoothness of cross-entropy can be achieved by the boundedness of input, but from the perspective of model capacity, such a result is not general. The function class here involves general functions (i.e., nonlinear mappings $\phi_j$), so for nonlinear models, smoothness of cross-entropy involves not only boundedness of the gradient of model function but also boundedness of second-order derivative. For deep networks, changes in parameters may cause drastic changes in second-order derivative, resulting in the norm of second-order derivative being unbounded. Hence, smoothness of cross-entropy is often difficult to hold. However, the Lipschitz continuity of cross-entropy is often established, and the boundedness of gradient of model function is guaranteed by various strategies in practice, e.g., input normalization, weight initialization, gradient clipping, and regularization. This implies the need to develop new theories and analytical methods for unbounded base losses. Under the new theoretical analysis method, tight bounds with no dependency on $c$ for cross-entropy loss can be obtained using its Lipschitz continuity, but improving the convergence rate of its bound wrt $n$ is still an urgent open problem to be solved. In the future, we will further study bounds with a faster convergence rate for unbounded Lipschitz base losses.
>
> **3. Response to C3**
>
> Since the output of MLL is a vector-valued function, we need to convert Rademacher complexity of the vector-valued class into complexity of a tractable scalar-valued class. For $\ell_2$ Lipschitz losses, the analysis of MLL can be traced back to a basic bound with a linear dependency on $c$ that comes from a typical inequality:
>
> $$\mathbb{E}\left[\sup_{\boldsymbol{f} \in \mathcal{F}} \frac{1}{n} \sum_{i=1}^n \sum_{j=1}^c \epsilon_{ij} f_j\left(\boldsymbol{x}_{i}\right)\right]$$
>
> $$\leq  c \max_j \mathbb{E}\left[\sup_{f_j} \frac{1}{n} \sum_{i=1}^n \epsilon_{ij} f_j\left(\boldsymbol{x}_{i}\right) \right].$$
>
> The dependency of bounds on $c$ can be improved to square-root. Such improvements essentially come from preserving the coupling among different components reflected by constraint $\\|\boldsymbol{w}\\| \leq \Lambda$.
>
> As a comparison, when $\\|\boldsymbol{w}_j\\|_2 \leq \Lambda$ for any $j \in [c]$,
>
> if we consider group norm $\\|\cdot \\|_{2, 2}$,
>
> we have $\\|\boldsymbol{w}\\|_{2, 2} \leq \sqrt{c}\Lambda$, which means that these improved bounds still suffer from a linear dependency on $c$. [1] improved the dependency of bounds on $c$ from linear to square-root in decoupling case for $\ell_2$ Lipschitz losses. We found that square-root dependency of bound in [1] on $c$ is inevitable for $\ell_2$ Lipschitz losses, which essentially comes from a $\sqrt{c}$ factor in the radius of empirical $\ell_2$ cover of projection function class. We also found that smoothness of base loss can eliminate the $\sqrt{c}$ factor, so that tight bound with no dependency on $c$, up to logarithmic terms, can be derived. In addition, according to the above core ideas, we combine smoothness of base loss with local loss space to develop novel local vector-contraction inequalities, thereby obtaining sharper bounds with a weak dependency on $c$ and a faster convergence rate wrt $n$. We have also explained proof processes, ideas and specific theoretical techniques in proof sketches, which mainly includes conversions between complexities of different classes and some lemmas required to achieve these conversions.
>
> [1] Generalization Analysis for Multi-Label Learning, ICML 2024.
>
> **4. Response to C4**
>
> We explain the multiplicative factor of 2, please refer to **Response 4 to Reviewer ZadH**.
>
> **5. Response to Literature**
>
> We give a detailed discussion of the paper pointed out, please refer to **Response 2 to Reviewer dgGN**.
>
> We will incorporate these discussions into the paper in an appropriate manner.
>
> **6. Response to Weakness**
>
> We will carefully check and revise the notation to ensure consistency, e.g., the definition of base losses $\ell_b$. $\widetilde{\Re}_{nc}(\mathcal{P}(\mathcal{F}))$ is worst-case Rademacher complexity of projection function class. We define worst-case Rademacher complexity in Definition 3.1,
>
> $\widetilde{\Re}_{nc}$ is the analog of the definition of worst-case Rademacher complexity for class $\mathcal{P}(\mathcal{F})$.

---

### Decision · Program_Chairs · 2025-05-01

**Decision:**

Accept (poster)

**Comment:**

This is a theoretical paper focused on multi-label classification. The authors derive generalization bounds for a specific class of classifiers, which depend only logarithmically on the number of labels.

The Reviewers are generally positive about the submission. However, they raised several concerns, including the limited model class and the specific class of loss functions to which the results apply, as well as the lack of references to related work (e.g., Busa-Fekete et al., 2022) and the unclear connection to results presented in (Zhang and Zhang, 2024). If accepted, the paper should be thoroughly revised to address these issues, incorporating the Reviewers' feedback and the clarifications provided by the Authors during the rebuttal phase.